# Rapid increase in summer surface ozone over the North China Plain during 2013–2019: a side effect of particulate matters reduction control?

Xiaodan Ma[1], Jianping Huang [2,6], Tianliang Zhao[1], Cheng Liu[3], Kaihui Zhao[4], Jia Xing[5], Wei Xiao[6]

[1]Collaborative Innovation Center on Forecast and Evaluation of Meteorological Disasters, Key Laboratory for Aerosol-Cloud-Precipitation of China Meteorological Administration, Nanjing University of Information Science and Technology, Nanjing 210044, China;

[2]I.M. System Group, Environmental Modeling Center, NOAA National Centers for Environmental Prediction, College Park, MD, USA

[3]Jiangxi Province Key Laboratory of the Causes and Control of Atmospheric Pollution/School of Water Resources and Environmental Engineering, East China University of Technology, Nanchang 330013, China

[4]School of Environment and Energy, South China University of Technology, Guangzhou 510006, China

[5]State Key Joint Laboratory of Environmental Simulation and Pollution Control, School of Environment, Tsinghua University, Beijing 100084, China

[6]Yale-NUIST Center on Atmospheric Environment, Nanjing University of Information Science and Technology, Nanjing, 210044, China

*Correspondence to:* Jianping Huang (jianping.huang@noaa.gov)

**Abstract.** While the elevated ambient levels of particulate matters with aerodynamic diameter of 2.5 micrometers or less ($PM_{2.5}$) are alleviated largely with the implementation of effective emission control measures, an opposite trend with a rapid increase is seen in surface ozone ($O_3$) in the North China Plain (NCP) region over the past several years. It is critical to determine the real culprit causing such a large increase in surface $O_3$. In this study, seven-year surface observations and satellite retrieval data are analyzed to determine the long-term change in surface $O_3$ as well as driving factors. Results indicate that anthropogenic emission control strategies and changes in aerosol concentrations as well as aerosol optical properties such as single-scattering albedo (SSA) are the most important factors driving such a large increase in surface $O_3$. Numerical simulations with National Center for Atmospheric Research (NCAR) Master Mechanism (MM) model suggest that reduction of $O_3$ precursor emissions and aerosol radiative effect accounted for 45 % and 23 % of the total change in surface $O_3$ in summertime during 2013–2019, respectively. Planetary boundary layer (PBL) height with an increase of 0.21 km and surface air

temperature with an increase of 2.1 °C contributed 18 % and 12 % to the total change in surface $O_3$, respectively. The combined effect of these factor was responsible for the rest change. Decrease in SSA or strengthened absorption property of aerosols may offset the impact of AOD reduction on surface $O_3$ substantially. While the MM model enables quantification of individual factor's percentage contributions, it requires further refinement with aerosol chemistry included in the future investigation. The study indicates an important role of aerosol radiative effect in development of more effective emission control strategies on reduction of ambient levels of $O_3$ as well as alleviation of national air quality standard exceedance events.

**1 Introduction**

Elevated ambient levels of ozone ($O_3$) are of great concern due to their important impact on human health, ecosystem productivity, atmospheric chemistry, and climate change (Monks et al., 2015; Tai et al., 2014; Tan et al., 2019). $O_3$ is produced by a series of photochemical reactions involving nitrogen oxides ($NO_x$ = NO + $NO_2$) and volatile organic compounds (VOCs) in the presence of solar radiation. Ambient levels of $O_3$ are highly dependent on emissions of $O_3$ precursors, solar radiation, and other physical processes such as regional and vertical transport (Sun et al., 2018; Ni et al., 2018; Liu et al., 2019; Wang et al., 2016b). While $O_3$ concentrations show a steady decreasing trend in Europe and North America, an opposite trend with an accelerating increase rate is observed in China (Lu et al., 2018; Li et al., 2019a). Due to high nonlinearity of $O_3$-$NO_x$-VOCs relationship and complexity of processes governing ambient levels of $O_3$, a large uncertainty remains in the determination of impact of different driving factors on changes in surface $O_3$ concentrations under the polluted atmospheric conditions. Thus, accurate quantification of relative contributions of individual factors to the large increase in surface $O_3$ concentrations over the heavily polluted regions such as China continues to represent one of major challenges to research communities and government policy makers.

Anthropogenic emissions are the key in driving change in surface $O_3$. With rapid development of industrialization and urbanization, anthropogenic emissions of $NO_x$ and VOCs, two major precursors of $O_3$ formation have been increasing significantly in China over the past several decades (Zeng et al.,2019). For instance, tropospheric columns of $NO_2$ (TCNO2), an indicator of anthropogenic emission intensity of $NO_x$ were increased by 307 % in Beijing from 1996 to 2011 (Huang et al., 2013), which caused a strong

increase trend of $O_3$ in the lower troposphere. Meanwhile, an increase in surface $O_3$ at a rate of 2 % $a^{-1}$

was observed in Beijing from 1995 to 2005 (Ding et al., 2007), and a similar increase with 1–2 ppb $a^{-1}$

was monitored at urban and remote sites in eastern China (Sun et al., 2016; Gao et al., 2017; Ma et al.,

2016; Tang et al., 2009). However, little light was shed on change in surface $O_3$ as compared to its

counterpart $PM_{2.5}$ which was elevated to the severe pollution level in eastern China especially over the

North China Plain (NCP) region (Zeng et al., 2019; Zhai et al., 2019). The severity of $PM_{2.5}$ pollution

has been largely alleviated after the stringent emission control strategies were implemented by Chinese

governments at national level in 2013 (Zeng et al., 2019). According to the estimate by Multi-resolution

Emission Inventory in China (MEIC), anthropogenic emissions of $PM_{2.5}$ decreased by approximately

%, $NO_x$ emissions decreased by 21 %, significant reductions were also seen in other air pollutants

such as $SO_2$ but not for VOCs which showed an increase of 2 % instead over the period of 2013–2017

(Zheng et al., 2018). As a result, monthly mean $PM_{2.5}$ concentrations decreased by 41 % for the Beijing–

Tianjin–Hebei (BTH) region which is similar to the NCP region presented in this study, and aerosol

optical depth (AOD) was reduced by 20 % in eastern China (Li et al., 2019a). However, an opposite

trend with an accelerating increase rate of $O_3$ was observed in the NCP region during this period (Lu et

al., 2018; Cooper et al., 2014). The fact that $O_3$ formation was dominated by VOC-sensitive regime may

partly account for such an increase in the NCP region, but it is not clear how much the change of surface

$O_3$ is attributed to anthropogenic emission control efforts.

    Aerosol radiative effect is another factor imposing a large constrain on change in surface $O_3$. Aerosols

attenuate surface-reached solar near-ultraviolet (UV) radiation effectively and reduce photolysis rate of

$NO_2$, a key parameter in determining $O_3$ formation. Impact of aerosol radiative effect on photolysis rate

of $NO_2$ or $O_3$ photochemical production is highly dependent on aerosol optical properties as described

by AOD, single-scattering albedo (SSA), and asymmetry factor. AOD is a measure of extinction of solar

beam by aerosols (e.g., dust and haze), used as a proxy of representing severity of fine particulate-matter

pollution or aerosol mass concentrations. SSA denotes the relative contributions of scattering versus

absorption effect to total aerosol extinction efficiency with "0" for pure absorption and "1" for pure

scattering effect. Both numerical simulations and observations showed that aerosols with UV-scattering

effect may accelerate photochemical production of $O_3$ but aerosols with strong absorption property (e.g.

mineral dust and soot) may inhibit $O_3$ production in the atmospheric boundary layer (Dickerson et al.,

1997; Mok et al., 2016). The lowest photolysis rate coefficient was closely linked with the highest AOD

(Liu et al., 2019; Dickerson et al., 1997). It was observed that surface $PM_{2.5}$ concentrations decreased by 41 % whereas surface $O_3$ increased at a rate of 3.1 ppb $a^{-1}$ over the BTH region from 2013 to 2017 (Li et al., 2019a). Decrease in $PM_{2.5}$ was considered as one of the important causes leading to such an increase in surface $O_3$ due to additional $O_3$ production associated with reduced sink of hydroperoxy radicals ($HO_2$) (Li et al., 2019a). They pointed out that increase in surface $O_3$ associated with decrease in $PM_{2.5}$ was

more prominent than that with reduction of $NO_x$ emissions over the NCP region where $O_3$ formation was dominated by VOC-limited regime. Liu and Wang (2020a, 2020b) found the reduction of PM emissions increased the $O_3$ levels by enhancing the photolysis rates and reducing heterogeneous uptake of reactive gases (mainly $HO_2$ and $O_3$), of which the latter is more important than the former. Similar impact associated with aerosol radiative properties on $O_3$ production was observed in other regions over the

world. For instance, the combined effect associated with optical properties of BrC and black carbon (BC) reduced the net change in $O_3$ production by up to 18 % as compared to BC alone in the Amazon Basin (Mok et al., 2016). Thus, surface $O_3$ changes are dependent on not only aerosol concentrations (AOD used as a proxy) but also aerosol optical properties such as SSA. Relative importance of different aerosol optical property parameters to change in surface $O_3$ needs to be addressed.

NCP, the largest alluvial plain of China, is surrounded by Mountains Yanshan with main peak of 2116 meters at the north, Mountains Taihang with the highest elevation of 2882 meters at the west, Mountains Dabie and Tianmu at the south, and bordered to Yellow Sea at the east (see Fig. 1). Such a complex terrain is not conducive to dispersion and dilution of air pollutants and makes them be trapped easily. Meanwhile, the total energy consumption was increased by more than five times from 1985 to 2016 (Zeng et al.,

2019). NCP has become one of the most polluted regions in China. The highest $PM_{2.5}$ concentration reached to $900 \mu g \cdot m^{-3}$ during winter and heavy $PM_{2.5}$ pollution events was the major concern to air quality during that period (Gu, 2013; An et al., 2019), but surface $PM_{2.5}$ concentrations have reduced substantially. Meanwhile, $O_3$ exceedance events became more frequent and more serious in the NCP region (Zhang et al., 2015; Lang et al., 2017; Zhai et al., 2019). Hourly surface $O_3$ concentrations went

up to 150.0 ppb and the increase rate reached to 3.1 ppb $a^{-1}$, much higher than those observed in other polluted regions such as Yangtze River Delta (YRD) and Pearl River Delta (PRD) in China (Li et al., 2019a; Lyu et al., 2019). The elevated surface $O_3$ has become an emerging critical air quality issue in this region (Wang et al., 2006; Shi et al., 2015). Understanding of the factors driving such as a rise in surface $O_3$ becomes a very hot topic (e.g., Li et al., 2019a; Li et al., 2019b). However, most of the related studies

are limited to the contributions of atmospheric chemistry and changes in $O_3$ precursors' emissions. Relative importance of aerosol radiative effect associated with substantial decrease in aerosols or $PM_{2.5}$ and meteorological variability to the enhancement of surface $O_3$ is not well qualified.

In this study, seven-year air quality observational data provided by the China National Environmental Monitoring Center (CNEMC) Network are examined to determine the temporal and spatial variations in

surface $O_3$ over the NCP region over the period of 2013–2019. A series of analyses are presented to investigate the long-term change trend of surface $O_3$ and the statistical relationships with $NO_x$ and VOCs emissions, meteorological variables, and aerosol radiative optical property parameters. A box model with Master Mechanism (MM) then is utilized to determine the response of surface $O_3$ to the key driving factors. The specific objectives include 1) to identify the key factors driving the increase in surface $O_3$

over NCP, the most polluted region in China; 2) to quantify the relative contributions of anthropogenic emissions (e.g., $NO_x$ and VOCs), aerosol concentrations, aerosol optical properties, and meteorological variability to the increase in surface $O_3$ in summertime during 2013–2019.

## 2 Data and Methods

### 2.1 Observational data

Data used in this study include hourly-averaged surface observations of $O_3$ and $PM_{2.5}$ from 2013 to 2019 provided by the CNEMC (http://106.37.208.233:20035/). UV data measured at the Yucheng site (i.e., YCA, 116.57º E, 36.87º N) in the NCP region are obtained from the Chinese Ecosystem Research Network (http://www.cern.ac.cn/) from years 2013 to 2016. AOD is derived from the monthly level-3 product of the Moderate Resolution Imaging Spectroradiometer (MODIS) instrument aboard the Aqua

satellite, reported at 550-nm wavelength with resolutions of 1° × 1° (Platnick, 2015). $TCNO_2$ data are retrieved from the daily level-3 products of the Ozone Monitor Instrument (OMI) aboard the Aura satellite with resolutions of 0.25° × 0.25° (Nickolay A. Krotkov, 2019). Short-wave radiation data are provided by Land Data Assimilation System (FLDAS) (NASA, 2018) at resolutions of 0.1° × 0.1°. SSA retrieved from OMI/Aura Near UV Aerosol Optical Depth and Single Scattering Albedo V003

(OMAERUV) (Torres, 2006) at 388 nm are used to evaluate the impact of aerosol scatting/absorption properties on change in surface $O_3$. Daily max temperature at 2 m ($T_{2max}$), 10 m wind speed and the planetary boundary layer height (PBLH) are derived from the Modern-Era Retrospective Analysis for

Research and Applications version 2 (MERRA-2) reanalysis data at horizontal resolutions of 0.5° × 0.625° (Global Modeling and Assimilation Office, 2015).

## 2.2 Model description and configurations

The MM model is utilized to quantify the relative contributions of anthropogenic emissions and aerosol optical and radiative properties to the change in surface $O_3$. The MM is a chemistry box model, originally developed and updated by the scientists at National Center for Atmospheric Research (NCAR). It includes a detailed and flexible gas phase chemical mechanism consisting of approximately 5000 reactions for simulating temporal variations in chemical species of interest. The hydrocarbon chemistry in the MM is treated explicitly with photo-oxidation of partly oxygenated organic species included. Alkanes, alkenes and aromatics are considered as initial hydrocarbon reagents in the gas-phase mechanism. The Gear-type solver is used in the MM model to handle so large numbers of chemical reactions and species and the integration time steps varied during the simulations (Madronich and Calvert, 1989). The TUV model is called by the MM model for update of chemical reaction rates every fifteen minutes. This model computes time-dependent chemical evolution of an air parcel initialized with a known composition and additional emissions. It is assumed that no dilution is included in the simulations given the difficulty of getting inputs to calculate the dilution rate. The transport in and out of air pollutants reached a quasi-equilibrium state over the study domain and no heterogeneous processes were included in the MM model. The MM model has been widely used to investigate impact of different factors such as emissions, chemistry, and meteorological conditions on simulations of $O_3$ and other chemical species (e.g., Liu et al., 2019; Geng et al., 2007).

Photolysis rate j(NO2) is calculated by using the Tropospheric Ultraviolet and Visible (TUV) radiation model which is embedded into the NCAR MM (Madronich S., 1999). In the fully-coupled system, the TUV is called by the MM model for update of photolysis rates of $NO_2$ and other chemical species (e.g., $H_2O_2$, $O_3$, $NO_3$, $N_2O_5$) every 15 minutes dynamically. The TUV model is initialized with the monthly means of AOD, SSA, and total columns of $O_3$ retrieved from satellite measurements as well as other meteorological parameters such as cloud fractions at the central point of NCP (36° N, 117.5° E) in June.

$HO_2$ radicals are important to $O_3$ formation. HONO photolysis as the primary production of OH radicals and formaldehyde (HCHO) photolysis as the net radical source of $HO_2$ can lead to major changes

in the HOx and NOx budget that may have an important effect on $O_3$ production and loss (e.g., Aumont et al., 2003; Brasseur et al., 2006; Lin et al., 2012a). The role of $HO_2$ radicals can be determined by the following reactions.

$$HO_2 + NO \rightarrow NO_2 + OH, \qquad\qquad R1$$

$$NO_2 + hv \rightarrow NO + O(^3P), \qquad\qquad R2$$

$$O_2 + O(^3P) \rightarrow O_3, \qquad\qquad R3$$

where $hv$ represents ultraviolet radiation at the wavelengths of 200–400 nm. The MM model has a capability of quantifying the role of radicals in $O_3$ formations under different pollution conditions.

The MM simulations are conducted for the predefined box as shown in Fig. 1 to represent ensemble mean behaviors and responses of the model to changes of different model inputs over the NCP region. The 24-hr simulations are conducted with the initial hour at 00z local time (LT). The inputs of the simulations include meteorological data (e.g., air temperature, cloud, and PBLH), aerosol radiative properties (i.e., AOD and SSA), and emissions. While all the meteorological inputs are generated from observational data, the initial values of chemical species such as VOC species (e.g., Acrylic, 2-methylbutane, Toluene, P-xylene, Isoprene), $N_2$, $O_2$, $H_2O$, $NO_2$, $O_3$ etc. are obtained from climatology or background values.

Emissions ($NO_x$ and VOCs) are calculated from the MEIC emission inventory. Aerosol radiative property parameters from MODIS and OMAERUV are assumed as constants for all the simulations. All the simulations are driven by the monthly means averaged over the entire NCP region. The temporal variations at an interval of 4 hours are included in the model inputs to represent the diurnal variations in different meteorological variables such as $T_{2max}$ and PBLH from MERRA-2 reanalysis.

Six groups with a total of sixteen numerical experiments with the MM model are designed to quantify the roles of different factors in driving change in $O_3$ concentrations (Table 1). Case A stands for the base that the emissions were generated from MEIC in base year 2012 (http://www.meicmodel.org/) with an adjustment for year-2013 use, and the spatial distributions of $NO_x$ and VOCs are presented in Fig. S1. Case B represents a scenario for year 2019 with $NO_x$ and VOCs emission changes by $-35\%$ and $+10\%$ with respect to the case in year 2013, respectively. The changes in $NO_x$ emissions ($-35\%$) and VOCs emissions ($+10\%$) in 2019 were obtained by extrapolating their respective changes during the period from 2013 to 2017 (Li et al., 2019a). Case C1 denotes a scenario with a decrease in AOD from 1.0 (i.e.,

the case for year 2013) to 0.75 (i.e., for year 2019) according to MODIS measurements and other six members in group C are used to examine the impact of varying AOD on the change in surface $O_3$. Case D1 is the one with a change of SSA from 0.95 (year 2013) to 0.93 (year 2019). Case E is a scenario with $T_{2max}$ increase from 29.9 °C in 2013 to 32.0 °C in 2019 based on regional average calculated with the MERRA-2 reanalysis in the NCP region. Case F is designed to assess the impact of the increased PBLH (i.e., increase from 0.76 km in 2013 to 0.97 km in 2019) on surface $O_3$ change in the NCP region. Case G is for the situation mimic for year 2019 representing the combined effect of changes in emissions, AOD, SSA, $T_{2max}$, and PBLH. 24-hr simulations are completed for each case to quantify the contributions of individual factors to the changes in surface $O_3$ from 2013 to 2019. More details of the numerical experiments are presented in Table 1.

## 3 Results and Discussion

### 3.1 Spatiotemporal variations in surface $O_3$, $PM_{2.5}$, AOD, and $TCNO_2$

Figure 2 shows a comparison of spatially distributed monthly means of the maximum daily 8-h average (MDA8) $O_3$, 24-h average $PM_{2.5}$, AOD, and $TCNO_2$ over the eastern China in June between 2013 and 2019 derived from in-situ and satellite observations. It is clear that NCP was the most polluted region with the highest values of MDA8 $O_3$, $PM_{2.5}$, AOD, and $TCNO_2$ over the past decade. 24-h average $PM_{2.5}$ concentrations were higher than 75.0 $\mu g\, m^{-3}$ (the Grade II National Ambient Air Quality Standard, NAAQS defined for residential areas) at most of the monitoring stations across the NCP region in June 2013. The highest 24-h average $PM_{2.5}$ reached to 766.0 $\mu g\, m^{-3}$ and the corresponding AOD was 1.0. As compared to the well-established monitoring network of $PM_{2.5}$, observational sites for $O_3$ were pretty sparse except for the BTH, YRD, and PRD across the eastern China in 2013. While the $TCNO_2$ was 2 times higher than that observed in North America (Stavrakou et al., 2008), the exceedance events of the MDA8 $O_3$ were not frequently observed across eastern China in 2013. $PM_{2.5}$ was the major air pollutant in the NCP region during that time period.

$PM_{2.5}$ concentrations, AOD, and $TCNO_2$ have been reduced substantially as a result of the implementation of strict anthropogenic emission reduction policy in 2013. For instance, monthly mean of $PM_{2.5}$ concentrations decreased from 95.5 $\mu g\, m^{-3}$ to 33.2 $\mu g\, m^{-3}$ with a percentage reduction of 65 %. Monthly mean AOD was reduced from 1.0 in 2013 to 0.75 in 2019, indicating that $PM_{2.5}$ continued

to decrease at a rate of $-10 \sim -11\%\ a^{-1}$ which was similar to that during 2013–2017 (Li et al., 2019a).

A similar decrease trend was seen in both $TNO_2$ (Fig. 2g-h) and in-situ $NO_2$ measurements (Fig. S2). On

the other hand, a rapid increase in surface $O_3$ concentrations was observed in the NCP region over the

past several years. The hot spots with the MDA8 $O_3$ higher than 75.0 ppb were extended to the entire

NCP as well as the neighbor regions in 2019 (Fig. 2b). The highest MDA8 $O_3$ reached to 112.8 ppb in

2018, which was even higher than the level (110.0 ppb) observed in Los Angeles (Lin et al., 2017). As

compared to the cases observed in 2017 (Li et al., 2019a; Li et al., 2019b), air pollution events with

higher surface $O_3$ became more severe and more frequent. The frequency of NAAQS exceedance events

for surface MDA8 $O_3$ (i.e., greater than 160 $\mu g\ m^{-3}$) in June increased from 30 % in 2013 to 63 % in

2019. Here percentage represents the proportion of MDA8 exceedance days to a total of 30 days (i.e.,

June).

    Reduction in $NO_x$ emissions and slight increase in VOC emissions could be part of the reasons causing

such an increase over the NCP region where $O_3$ formation was dominated by VOC-limited regime. To

better understand the relationship of increase in surface $O_3$ with the decrease in $NO_2$, the change in

monthly mean Ox (a sum of $O_3$ and $NO_2$) was plotted in Fig. S3. It is clear that Ox showed an increasing

trend over the past 7 years during daytime and nighttime in both urban Beijing and the NCP region.

Meanwhile, Li et al. (2019a) attributed the increase to aerosol chemistry that removal of $HO_2$ radicals

was reduced and more $O_3$ production was promoted. On the other hand, attenuation of UV radiation

became less evident as $PM_{2.5}$ or AOD continually decreased. Strengthening UV radiation may accelerate

photolysis of $NO_2$ and eventually led to more $O_3$ production. Importance of aerosol radiative effect in the

increase in surface $O_3$ via the way of accelerating photolysis of $NO_2$ can be further evaluated through

numerical experiments.

    Meteorological conditions are another critical factor affecting $O_3$ production. Typically, higher air

temperature is responsible for higher photochemical reaction rates and more $O_3$ photochemical

production (Porter and Heald, 2019). As shown in Fig. 3, NCP was the hottest spot region with $T_{2max}$

which was about 4.0 °C higher than that in the neighbor regions. In addition, increase rate of $T_{2max}$ in the

NCP was larger than that observed in other regions in eastern China. $T_{2max}$ and surface-reaching

shortwave radiation increased by 3 % and 7 %, respectively, over the past several years. In addition to

man-made factors such as urbanization and industrialization, decrease in aerosols (e.g., $PM_{2.5}$ and AOD)

could be an important factor driving such a rise in air temperature due to weakening aerosol radiative

effect.

**3.2 Yearly changes in surface O$_3$ during 2013–2019 and driving factors**

As presented above, NCP was the most polluted region with extremely high ambient levels of air pollutants. Surface O$_3$ showed a rapid increase over the period from 2013 to 2019 while PM$_{2.5}$ and other pollutants such as NO$_x$ experienced a significant reduction. O$_3$ has become a major air quality concern

in summer. June was the month with the highest monthly mean MDA8 O$_3$ concentrations (Fig. S4). In this section, we attempt to investigate the yearly change rate and to identify the factors that drove such a large increase in surface O$_3$ over the NCP region throughout the period of 2013-2019.

Figure 4 shows the yearly changes in monthly means of MDA8 O$_3$, PM$_{2.5}$, AOD, SSA, TCNO$_2$, T$_{2max}$, and PBLH over the NCP region in June from 2013 to 2019. The change in monthly mean of surface

MDA8 O$_3$ showed an opposite trend to that of PM$_{2.5}$ concentrations and other air pollutants. A similar large change trend was seen in diurnal variation patterns (Fig. S5). The increase rate of monthly mean MDA8 O$_3$ (4.6 ppb a$^{-1}$) during 2013–2019 was much higher than that observed in the same region during the period of 2005–2015 (1.1 ppb a$^{-1}$) (Ma et al., 2016), and other regions such as Mountain Tai, YRD, Hong Kong, and North America where the changes were less than 2.1 ppb a$^{-1}$ during the similar time

period (e.g., Sun et al., 2016; Gao et al., 2017; Wang et al., 2017; Xu et al., 2019). At the same time, a large decrease can be found from the time series of PM$_{2.5}$, AOD, TCNO$_2$ (Fig.4b–d), and in-situ NO$_2$ measurements (Fig. S6). It is noted that SSA also showed a decreasing trend (Fig. 4e). Decrease in SSA was likely due to the fact that reduction of inorganic aerosols (e.g., sulfate and nitrate) was larger than that of carbonaceous ones (Zhang et al., 2020). Another noticed feature is that MDA8 O$_3$ showed a

decreasing trend in 2019 relative to 2018, which was opposite to that during 2013–2018 (Fig. 4a). It is worth to witness the change trend in the coming years.

To understand the factors driving the change in surface O$_3$, a series of scatter plots are presented to examine the relationships between the surface MAD8 O$_3$ and individual factors such as aerosol optical properties (i.e., AOD and SSA), TCNO$_2$, T$_{2max}$, and surface-reaching short-wave radiation over the past

seven years in June (Fig. 5). The values discussed here represent the monthly means. MAD8 O$_3$ showed two different regimes with an opposite dependence of O$_3$ formation on PM$_{2.5}$ concentrations. The first regime showed a decrease trend with increasing surface PM$_{2.5}$ when PM$_{2.5}$ concentrations were less than

approximately 140.0 μg $m^{-3}$ whereas the second one showed no trend with increasing PM2.5 when

PM2.5 concentrations were higher than 140.0 μg $m^{-3}$. The first regime was highly related to aerosol

radiative effect, which has been discussed above. For the 2nd regime, the impact of aerosol radiative effect

on surface O3 photochemical production seemed very minor or even negligible. Instead, O3 production

was suppressed significantly and MAD8 O3 concentrations were less than 20.0 ppb. In this case, removal

of surface O3 through titration of NO was not effective and surface O3 showed an increase rather than a

decrease trend with increasing NOx concentrations under the strong NOx conditions as indicated by

TCNO2 higher than 40–45× $10^{15}$ ($cm^{-2}$) in the troposphere. Here the threshold value of 140.0 μg $m^{-3}$

represents an observed reality in this region but it needs to investigate whether such a threshold value

exists in other regions.

Figures 5d–e further demonstrate the critical role of meteorological factors in change of surface O3.

MDA8 O3 showed a near linear increasing trend with increasing T2max and surface-reaching shortwave

radiation with respective linear regression correlation coefficients of 0.88 and 0.93. Increase in T2max and

strengthening shortwave radiation caused by decrease in PM2.5 (a proxy of aerosols) played a positive

role in driving the increase in surface O3 in the NCP region. On the other hand, MDA8 O3 showed a

decrease trend with 10–m wind speed (Fig. 5f). That may explain why improvement of stagnation

atmospheric conditions may alleviate severity of surface O3 pollution to some extents. The positive

correlation between the PBLH and O3 shown in Fig. 5g represents one case when radiation is stronger

and temperature is higher, that are favorable for O3 formation. Meanwhile, higher PBLH could enhance

the transport down of O3-enriched air aloft, resulting in an increase in surface O3 (Reddy et al., 2012).

On the other hand, some studies found a negative correlation between the PBLH and O3. They claimed

that a shallower PBL may suppress the dispersion of pollutants and lead to higher O3 (Yan et al., 2018;

Jiang et al., 2016; Wei et al., 2016; Huang et al., 2005)

Enhancement of UV radiation resulting from reduction in surface PM2.5 represents one of important

mechanisms in driving increase in surface O3 concentrations. It can be further illustrated by Fig. 6. While

UV radiation displays a nonlinear decreasing trend with surface PM2.5 concentrations, surface O3 (hourly)

shows a near linear increasing trend with surface-reached UV radiation. UV radiation attenuation

approaches to a constant with a value of 0.1–0.3 MJ $m^{-2}$ when surface PM2.5 concentrations reach to

around 300 μg $m^{-3}$ or above.

Analyses presented above demonstrate that all the exceedance events of MDA8 are observed under

conditions with PM$_{2.5}$ less than 60 μg $m^{-3}$, TCNO$_2$ of equal to or less than $5.0 \times 10^{15}$ (cm$^{-2}$), T$_{2max}$ higher than 28.0 °C, and surface-reaching shortwave radiation stronger than 250.0 W m$^{-2}$. Reduction in aerosols (e.g., surface PM$_{2.5}$ as a proxy) concentrations may strengthen UV radiation, increase T$_{2max}$, and eventually promote more surface O$_3$ production.

**3.3 Relative contributions of different driving factors to increase in surface O$_3$**

In this section, the box model MM is utilized to quantify the relative contributions of individual driving factors to the increase in surface O$_3$ over the NCP region during 2013–2019. A simulation-observation comparison is presented to evaluate the performance of the MM model on simulations of surface O$_3$ (Fig. 7), of which the O$_3$ observations averaged over all the stations in NCP is considered as the standard observed concentrations. The simulated O$_3$ peak was about one hour later than the observation, which was likely due to uncertainty of emission inventory and other meteorological factors. Overall, the MM model was able to mimic the observed variation pattern and peak value as indicated by the correlation coefficient of 0.95 between simulated and observed O$_3$.

A series of numerical experiments were then completed with the MM model to quantify the relative contributions of anthropogenic emissions (i.e., NO$_x$ and VOCs), AOD, SSA, air temperature, and PBLH to the change in surface O$_3$ over the NCP region during 2013–2019. The results are presented in Table 2. The changes in emissions of O$_3$-precursors (i.e., NO$_x$ and VOC$_s$) (i.e., Case B) and decrease of AOD (i.e., Case C1) were the two major contributors with their respective positive contributions of 45 % and 70 % to the increment in surface O$_3$. But increase in surface O$_3$ associated with AOD reduction was largely offset by the reduction in SSA. Moreover, air temperature played a non-negligible role and the increase in T$_{2max}$ accounted for 12 % of surface-O$_3$ enhancement (Case E). Meanwhile, the increase of PBLHs also contributed about 18 % to the increment in surface O$_3$ (Case F). As indicated by Case G, the combined effect by multiple factors was larger than the simple summation of individual factor's contributions or the total percentage contributions by individual factor was less than 100 %. This is likely due to the fact that O$_3$ production is not the linear function of individual factor's contribution. Complex interplay among different factors may account for rest of the increase (i.e., 2 %).

It is not surprised that reduction in NO$_x$ emissions brought about increase in surface O$_3$ since O$_3$ formation was dominated by VOC-limited regime in most parts of the NCP region. Several numerical

experiments were conducted to understand the mechanism of reduced $PM_{2.5}$ or AOD facilitating the increase in surface $O_3$. It is known that photolysis rate of $NO_2$, $j(NO_2)$ plays a critical role in $O_3$ formation. Parameter $j(NO_2)$ was highly dependent on aerosol optical properties such as AOD and SSA, as well as solar zenith angle ($\theta$) (Dickerson et al., 1997). As shown in Fig. 8a, decreasing AOD was conducive to

photolysis of $NO_2$ due to reduction of attenuated UV radiation entering the PBL. However, weakened scattering or strengthened absorption property of aerosols (i.e., reduced SSA) may attenuate the UV entering the PBL, deaccelerating photolysis of $NO_2$. Thus, decrease in SSA may counteract the impact associated with decrease in AOD, which may slow down the increase in surface $O_3$ to some extents. In addition, $j(NO_2)$ showed the highest value at noontime ($\theta = 0°$ or $\sec\theta = 1$) and tended to decrease

when $\theta$ became larger (i.e., early morning or late afternoon). Figure 8b further demonstrates that $O_3$ formation or MDA8 $O_3$ showed a near linear increasing trend with $j(NO_2)$. While decrease in $PM_{2.5}$ concentrations or AOD strengthened the UV amount entering the PBL reduction in SSA may counteract impact of decreased AOD partially. But impact of AOD outpaced that of SSA. Thus, surface $O_3$ (e.g., MDA8 $O_3$) still showed a large increase with the combined effect of AOD and SSA over the past several

360     years.

Now let us turn our attention to $O_3$-chemistry in the varying polluted region. As illustrated in Fig. 9, $HO_2$ radicals were sensitive to aerosol properties (i.e., AOD and SSA) but the sensitivity was highly relied on the solar zenith angle ($\theta$). $HO_2$ radical was more sensitive to AOD or SSA in the afternoon than in the morning while photolysis rate of $HO_2$ is more sensitive to AOD or SSA. It is noted that higher net

$O_3$ production is highly associated with the faster decrease in $J(O_3)$ than $J(NO_2)$ in the afternoon (Gerasopoulos et al., 2006). $HO_2$ radical abundance reduced as aerosol optical property became more absorptive. This indicates that decrease in SSA may cause reduction of $HO_2$, less $NO_2$, and then less $O_3$ production. The $HO_2$ peak hour was matched well with that of $O_3$ peak (around 15 p.m. LT), further confirming its important role in $O_3$ formation. Decrease in AOD may accelerate production of $HO_2$

radicals or slow down their sink, which was conducive to production of $NO_2$ (Li et al., 2019a) but decrease in SSA may offset its impact if aerosols show strong absorption property. Meanwhile, strengthened UV associated with weakened aerosol radiative effect was conducive to photolysis of $NO_2$. As a result, more $O_3$ is produced. This accounted for substantial increase in surface $O_3$ while $PM_{2.5}$ decreased over the past several years (2013 to 2019). The results are consistent with the finding by Li et

al. (2019a).

**4 Discussions**

In this study, a box model NCAR MM with the detailed $NO_x$-VOC-$O_3$ chemistry included is utilized to quantify percentage contributions of emissions, aerosol optical properties, and meteorological variabilities to increase in surface $O_3$ over the NCP region during 2013–2019. The findings may provide more scientific evidence to policy makers on developing more effective control strategies on reduction in ambient levels of $O_3$ as well as exceedance events. However, several points deserve further discussions.

First, the impact of aerosol radiative effect on surface $O_3$ formation is dependent on not only aerosol abundance (i.e., AOD) but also aerosol scattering/absorption property (i.e., SSA). Their impacts can be offset to some extents when AOD and SSA show the same change trend (either increase or decrease) or can be strengthened substantially when both AOD and SSA show an opposite change trend. Here the study on the NCP region represents the first case since both AOD and SSA showed a decrease trend over the past several years. Even so, the combined impact of aerosol radiative effect due to reductions in AOD and SSA still contributed 23% of the total change in surface $O_3$ in the NCP over the past several years. This reminds us that the impact of aerosol radiative effect could be more substantial if both AOD and SSA show an opposite change trend. Moreover, as compared to impact of change in AOD on surface $O_3$ formation (e.g., Dickerson et al., 1997; Wang et al., 2016a; Xing et al., 2015; Xing et al., 2017), studies on impact of change in SSA on surface $O_3$ formation are fewer (Dickerson et al., 1997; Mok et al., 2016). Thus, changes of individual aerosol radiative property parameters must be addressed carefully in order to present more accurate quantification of impact of aerosol radiative effect on change in surface $O_3$.

Second, the MM model does not include aerosol chemistry. As presented above, the MM model as a box model with the detailed $O_3$-$NO_x$-VOCs relationship allows us to quantify relative contributions of individual factors to increase in surface $O_3$. Overall, the model results are comparable to those by using three-dimensional (3D) chemistry and transport models (CTMs) (e.g., Liu and Wang 2020a, 2020b). For instance, the MM model result indicates that 45 % of increase in surface $O_3$ was attributed to reduction of anthropogenic emissions of $NO_x$ in the NCP region during 2013–2019, which fell in the range of the results with 3D CTM modeling. Among the 3D modeling studies, Li et al. (2019a) found that anthropogenic emissions contributed about 10 % of change in surface $O_3$ in summertime from 2013 to 2017 and Sun et al. (2019) showed the percentage contribution of anthropogenic emissions was 63 % over the eastern China. However, there is some substantial difference between the MM model result and

that of 3D CTMs in terms of percentage contribution of aerosol radiative effect to changes in surface $O_3$.

The MM model showed that aerosol radiative effect was ranked as the $2^{nd}$ contributor to the change in

surface $O_3$ in this region. The percentage contribution was larger than that presented by other studies (Li

et al., 2019a; Xing et al., 2015). This is partly because this study is focused on the impact on MDA8 $O_3$

whereas their studies investigated the impact on diurnal variations of surface $O_3$. In addition, Li et al.

(2019a and 2019b) and Liu and Wang (2020a and 2020b) pointed out that aerosol chemistry played the

most important role in the enhancement of surface $O_3$ in this region through modification of $HO_2$ radicals

that produce additional $O_3$ formation. However, the MM model does not include aqueous-phase

chemistry that has been implemented in the 3D meteorology/chemistry models (e.g., Li et al., 2019a; Liu

and Wang, 2020a, 2020b), which could be another possible reason in response to such a difference. Thus,

inclusion of detailed aerosol chemistry and observation-based uptake coefficients in a box model like

MM is necessary to provide more accurate assessment of impact of aerosol radiative effect on surface $O_3$

change.

Third, as compared to 3D meteorology/chemistry coupling model(s), box model does not include

complex physical processes such as regional transport, vertical transport, and cloud formation, etc. The

influence of changing meteorological factors on the change trend in surface $O_3$ may vary greatly with

regions and time. In addition to air temperature and the boundary layer conditions, other meteorological

factors such as cloud cover, precipitation, wind fields played an important role in driving the changes in

surface $O_3$ observed in many places of China (Liu and Wang, 2020a). Computational resource and

workload that a box model requires are much less than that a 3D chemical transport model needs. This

may allow us to complete a series of designed numerical experiments to quantify the roles of individual

factors easily with limited computational resources. It is acceptable by using a box model if terrains are

relatively flat in the box, horizontal gradients of emissions and air pollutant concentrations are not strong,

and transport in and out reaches a relative equilibrium state. As shown in Fig. S1 and Fig. 2, the NCP

region defined in this study represents the most polluted part in eastern China, anthropogenic emissions

appear to distribute relatively uniform across the region. To this extent, it is appropriate to examine $O_3$

formation and its response to changes of different factors such as emissions, meteorological conditions,

and aerosol radiative properties by using a box model in the NCP region. However, some other physical

processes such as long-range transport may exert an important impact on change in surface $O_3$ (e.g., Han

et al., 2018; Gaudel et al, 2018). It is reminded that the box model results present an ensemble-mean

behavior for the given box but need further evaluations by using a complex meteorology/chemistry coupling model such as Weather Research and Forecasting model with Chemistry (WRF/Chem).

     Forth, some other important factors may exert an important impact on surface $O_3$ concentrations, but they are not discussed in this study. Stratospheric intrusion and change in tropospheric $O_3$ could exert an important impact on $O_3$ in the atmospheric boundary layer (ABL) and near surface. For instance, Jiang

et al. (2015) presented a factor analysis on an $O_3$ episode observed in the southeast costal of China and found that the downward transport of $O_3$ from the UTLS region driven by a typhoon was the key factor causing a large increase in surface $O_3$ by 21-42 ppb. Thus, the impact of tropospheric $O_3$ should be taken into account when the appropriate observational data are available in NCP region. Another factor is synoptic patterns. As an example, high concentrations of surface $O_3$ or $O_3$ episodes occurred in western

Mediterranean and central Europe were usually linked with anticyclone synoptic pattern which led to a large-scale subsidence, clear sky, and high temperature (e.g., Kalabokas et al., 2013; Kalabokas et al., 2017). In addition, Yin et al. (2019) found that synoptic patterns played a critical role in summer $O_3$ pollution events in eastern China. Under the control of zonally enhanced East Asian deep trough, the local hot, dry air and intense solar radiation enhanced the photochemical reactions and produced more

$O_3$. The inter-annual magnitude variations of the domain synoptic patterns may have an important impact on surface $O_3$, and its impact on the long-term change in surface $O_3$ needs further investigation.

**5 Summary and conclusions**

In this study, seven-year long surface observational air quality data are presented together with satellite retrieval measurements of $TCNO_2$, AOD and SSA to investigate long-term change trend of surface $O_3$

over the NCP region in summer from 2013 to 2019. A comprehensive statistical analysis is completed to explore the relationship of MDA8 $O_3$ with $PM_{2.5}$ concentrations, tropospheric columns of $NO_2$, AOD, and meteorological variables such as $T_{2max}$, surface-reaching shortwave radiation, wind speed, and PBLH. A box model representing the $O_3$-$NO_x$-VOCs relationship is then utilized to quantify the relative contributions of different driving factors to the increase in surface $O_3$ in the NCP region over the period

of 2013–2019.

     The observational analysis indicates, while $PM_{2.5}$ concentrations continued to decrease with a rate of 9.5 $ug\ m^{-3}\ a^{-1}$, surface $O_3$ showed an accelerated increase trend at a rate of 4.6 ppb $a^{-1}$ over the NCP

region during summertime from 2013 to 2019. Both decrease in $PM_{2.5}$ and reduction in $NO_2$ are the two key factors leading to such an increase in surface $O_3$. The former is closely associated with the attenuation of UV entering the PBL whereas the latter is related to the fact that $O_3$ photochemical production in the NCP region is dominated by VOC-limited regime. The trend analysis of satellite retrieval measurements revealed an obvious increase in $T_{2max}$ at the rate of 0.34 $^oC$ $a^{-1}$, a rapid decrease in AOD from 1.0 in 2013 to 0.75 in 2019, and a reduction in SSA from 0.95 to 0.93. The changes of both $T_{2max}$ and AOD were conducive to photochemical production of $O_3$ whereas the variability of aerosol scattering/absorption property (i.e., decrease in SSA) may largely offset the impact of AOD reduction.

The sensitivity studies with the box model MM indicate that reduction of emissions (i.e., $NO_x$), meteorological conditions, and aerosol radiative effect associated with decrease in aerosol concentrations were the three most important factors in driving such a large increase in surface $O_3$. They accounted for 45 %, 30 %, and 23 % of the total increase in surface $O_3$, respectively over the NCP region in summertime during 2013-2019. For the meteorological contribution, increases in the PBLH and air temperature (e.g., $T_{2max}$) were responsible for 18 % and 12 % of the total change of surface $O_3$, respectively. The percentage contribution of aerosol radiative effect (23 %) represented the net changes caused by aerosol concentrations (i.e., AOD) and aerosol radiative property (scattering/absorption, SSA) (70 % vs. − 47 %). The model results further demonstrated that decrease in SSA (i.e., more absorptive) may lead to reduction in $HO_2$ radicals and $NO_2$ concentrations, and then less $O_3$ production, which may largely counteract impact of aerosol radiative effect associated with decrease in AOD.

This study has a strong implication that development of more effective control strategies on surface $O_3$ reduction needs to consider impact of aerosol radiative effect as well as the change of aerosol scattering/absorption properties (i.e., AOD and SSA).

**Data availability:** Data used in this paper can be provided by Xiaodan Ma (xaiodanma_nuist@163.com) upon request.

**Author contributions:** JH came up with the original idea of this study. XM and JH designed the numerical simulations. XM conducted the data analysis and the first draft of manuscript and JH did the

edit work. TZ, CL, KZ, JX and WX were involved in the scientific interpretation and discussions. All

the authors commented on the paper.

**Competing interests:** The authors declare that they have no conflict of interest.

**Acknowledgments:** This research was supported by the National Key R&D Program Pilot Projects of China (2019YFC0214604), the National Natural Science Foundation of China (Grant no. 41575009, no. 41830965 and no. 91544109), the Postgraduate Research & Practice Innovation Program of Jiangsu Province (KYCX20_0924) and the Jiangxi Provincial Natural Science Foundation (20202BAB213019).

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

**Table 1.** A summary of numerical experiments with the NCAR MM model.

| Case | NOx emission | VOCs emission | AOD | SSA | $T_{2max}$ (°C) | PBLH (km) |
|------|------|------|------|------|------|------|
| A | 2013[*] | 2013[*] | 1.0 | 0.95 | 29.9 | 0.76 |
| B | **2019[+]** | **2019[+]** | 1.0 | 0.95 | 29.9 | 0.76 |
| C1 | 2013 | 2013 | **0.75** | 0.95 | 29.9 | 0.76 |
| D1 | 2013 | 2013 | 1.0 | **0.93** | 29.9 | 0.76 |
| E | 2013 | 2013 | 1.0 | 0.95 | **32.0** | 0.76 |
| F | 2013 | 2013 | 1.0 | 0.95 | 29.9 | **0.97** |
| G | **2019** | **2019** | **0.75** | **0.93** | **32.0** | **0.97** |
| C2 | 2013 | 2013 | **0.5** | 0.95 | 29.9 | 0.76 |
| C3 | 2013 | 2013 | **0.6** | 0.95 | 29.9 | 0.76 |
| C4 | 2013 | 2013 | **0.7** | 0.95 | 29.9 | 0.76 |
| C5 | 2013 | 2013 | **0.8** | 0.95 | 29.9 | 0.76 |
| C6 | 2013 | 2013 | **0.9** | 0.95 | 29.9 | 0.76 |
| C7 | 2013 | 2013 | **1.1** | 0.95 | 29.9 | 0.76 |
| C8 | 2013 | 2013 | **1.2** | 0.95 | 29.9 | 0.76 |
| C9 | 2013 | 2013 | **1.25** | 0.95 | 29.9 | 0.76 |
| D2 | 2013 | 2013 | 1.0 | **0.94** | 29.9 | 0.76 |

[*]Year 2013: $NO_x$ emission is $2.0 \times 10^{12}$ mole. $cm^{-2} s^{-1}$, and VOCs emission is $7.3 \times 10^{9}$ mole. $cm^{-2} s^{-1}$

[+]Year 2019: $NO_x$ emission is $1.3 \times 10^{12}$ mole. $cm^{-2} s^{-1}$, and VOCs emission is $8.0 \times 10^{9}$ mole. $cm^{-2} s^{-1}$.

**Table 2.** Relative percentage contributions of emissions (case B), AOD (case C1), SSA (case D1), air temperature (case E), and PBLH (case F) to the change in MDA8 $O_3$ over the NCP region during 2013–2019.

| | MDA8 $O_3$ (ppb) | Concentration Change (ppb) | Percentage Change (%) | Percentage Contribution (%) |
|---|---|---|---|---|
| A | 55.35 | | | |
| B | 59.25 | 3.90 | + 7 % | + 45 % |
| C1 | 61.46 | 6.11 | + 11 % | + 70 % |
| D1 | 51.22 | − 4.13 | − 7 % | − 47 % |
| E | 56.43 | 1.08 | + 2 % | + 12 % |
| F | 56.95 | 1.60 | + 3 % | + 18 % |
| G | 64.09 | 8.74 | + 16 % | |




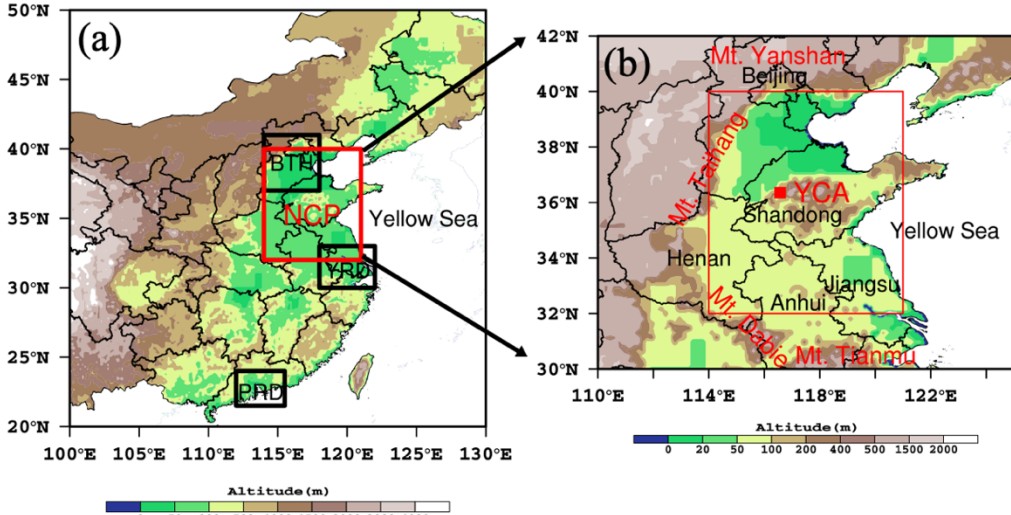

**Figure 1.** (a) Locations of North China Plain (NCP, 32°–40° N, 114°–121° E) and other three major air pollution regions, Beijing-Tianjin-Hebei (BTH, 37°–41° N, 114°–118° E), Yangtze River Delta (YRD, 30°–33° N, 118°–122° E), and Pearl River Delta (PRD, 21.5°–24° N, 112°–115.5° E) in China with terrain heights included and (b) location of ultraviolet (UV) radiation observational site, YCA (Yucheng site), areas covered by the NCP region and mountains surrounded.


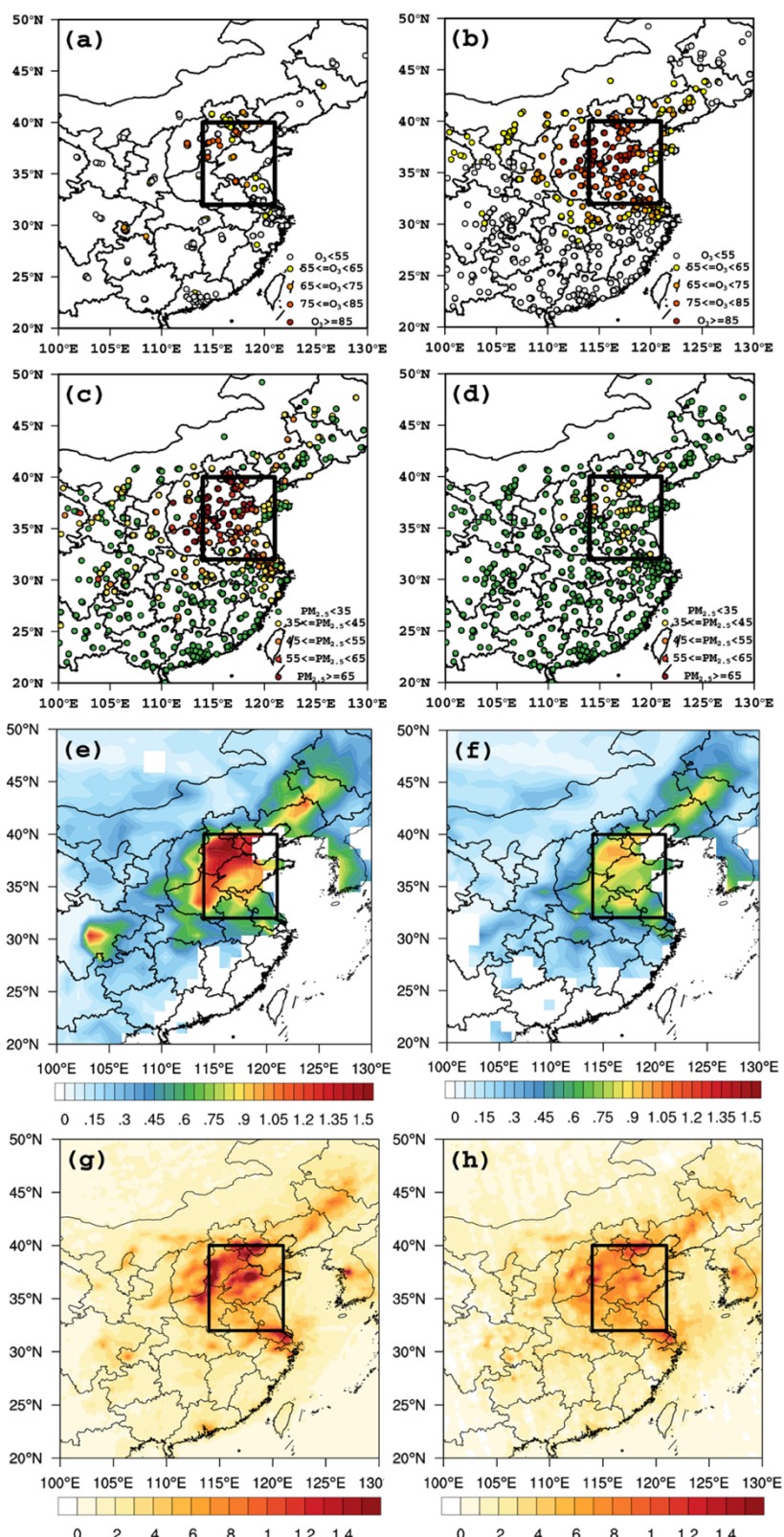

**Figure 2.** A comparison of spatial distributions of monthly mean of MDA8 O₃ (ppb) (a, b) and PM₂.₅ (μg $m^{-3}$) (c, d) obtained from in-situ observations, AOD (e, f) and tropospheric column of NO₂ (TCNO₂, $10^{16}$ cm$^{-2}$) (g, h) derived from satellite observations between 2013 (in left column) and 2019 (in right column) in eastern China (NCP indicated by the box).

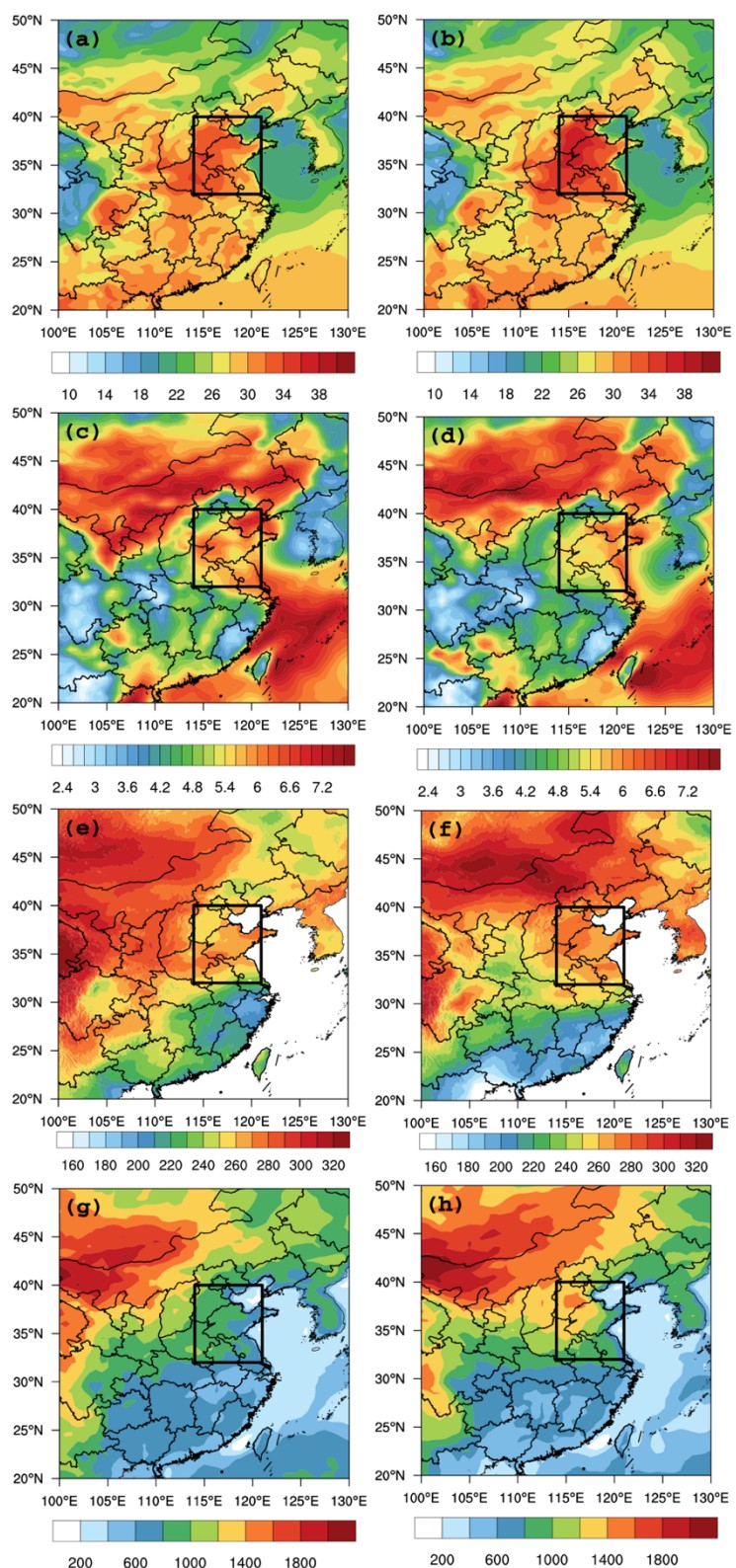

**Figure 3.** A comparison of spatial distributions of monthly mean of $T_{2max}$ (℃, a and b), wind speed (m s$^{-1}$, c and d), surface reaching short-wave radiation (W m$^{-2}$, e and f) and PBLH (m, g and h) between 2013 (in left column) and 2019 (in right column) in eastern China (the NCP indicated by the box).

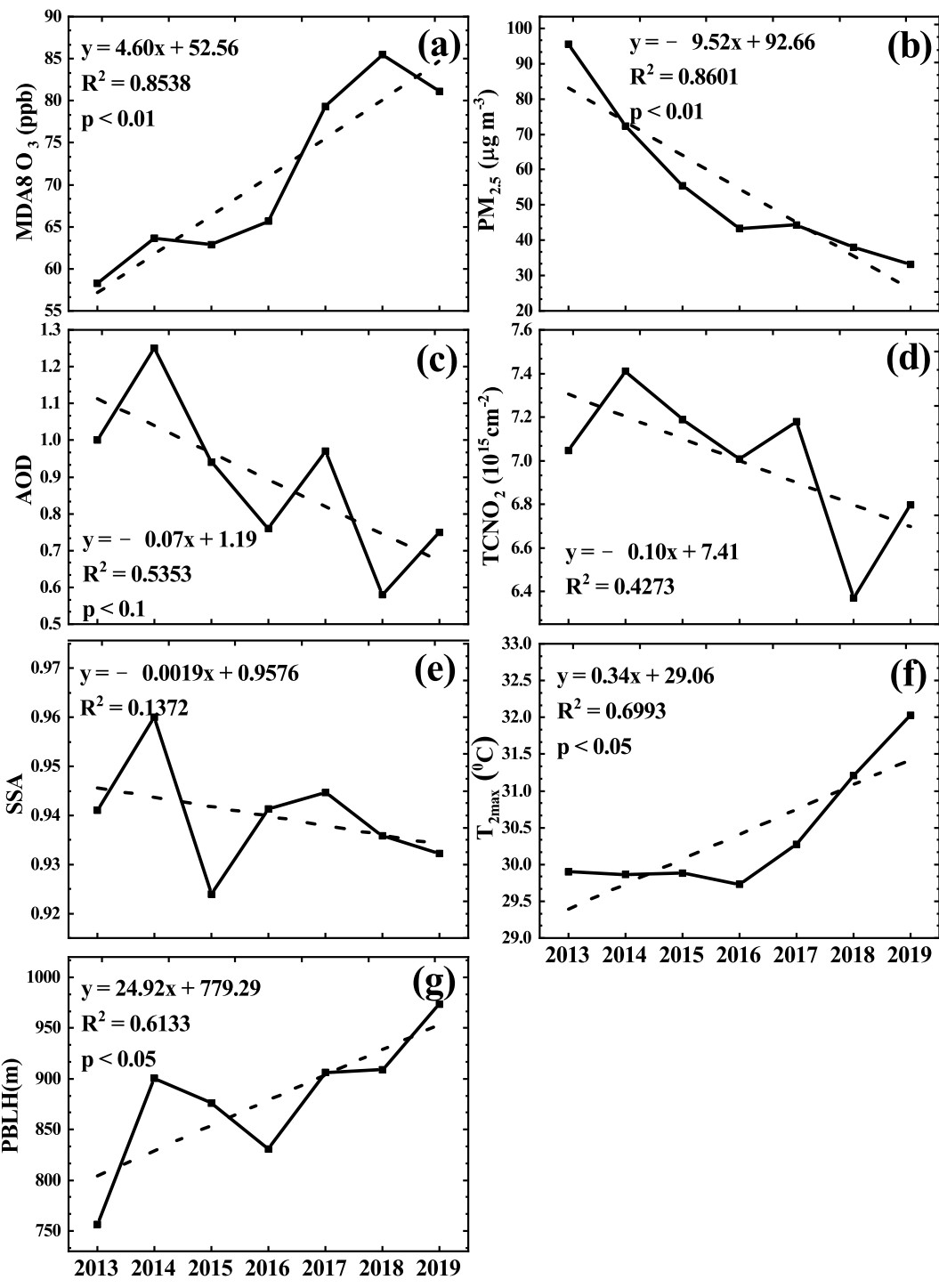

**Figure 4.** Long-term changes in monthly mean of (a) MDA8 $O_3$, (b) $PM_{2.5}$, (c) AOD, (d) $TCNO_2$, (e) SSA, (f)

$T_{2max}$, and (g) PBLH averaged over the North China Plain in June over the period of 2013–2019.


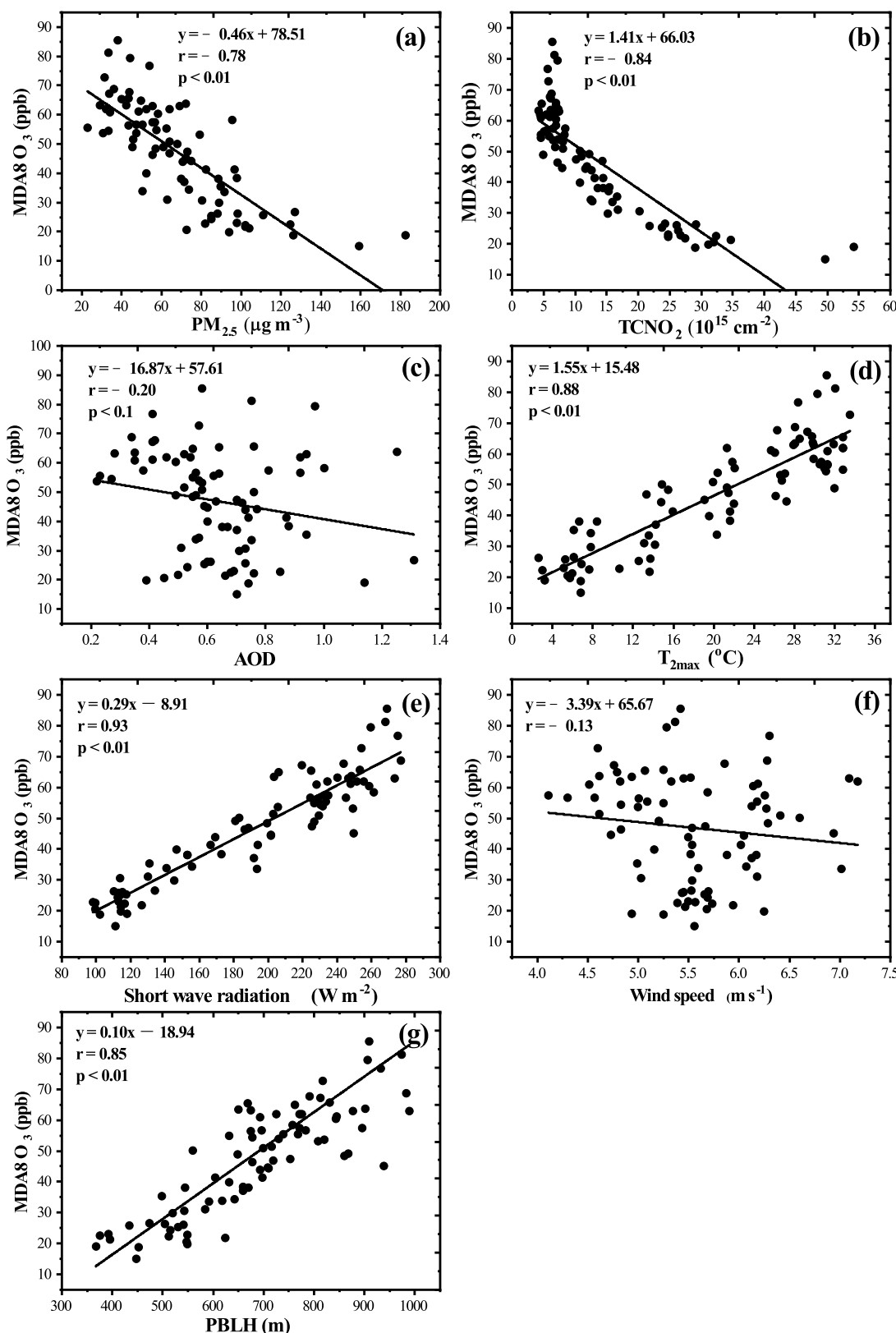

**Figure 5.** Response of MDA8 $O_3$ to (a) $PM_{2.5}$, (b) $TCNO_2$, (c) AOD, (d) $T_{2max}$, (e) shortwave radiation, (f) wind speed, and (g) PBLH observed in the NCP region, China during 2013–2019.

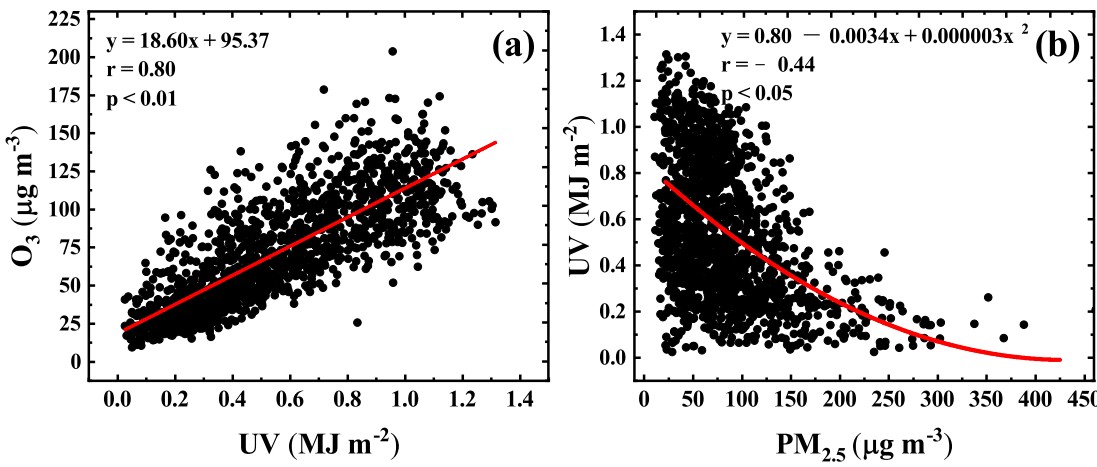

**Figure 6.** a) The relationships of surface $O_3$ concentrations (hourly) with (a) UV radiation and (b) UV radiation with

$PM_{2.5}$ concentrations based on the observations at Yucheng site during the time period of 08–17 LT in June, 2013–

2016.

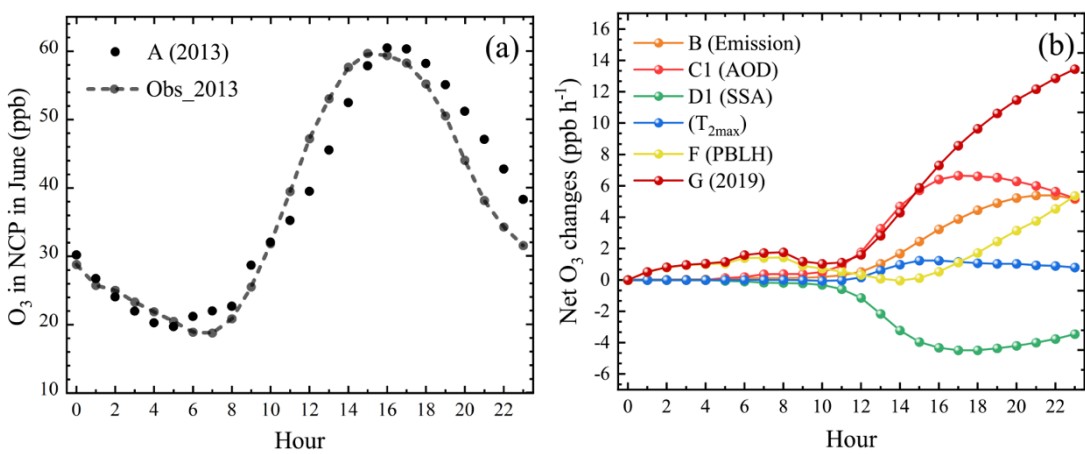

**Figure 7.** Comparisons of (a) regional averaged surface $O_3$ observations in NCP and simulated surface $O_3$ (A, control case) and (b) simulated net changes in $O_3$ among different driving-factor conditions.

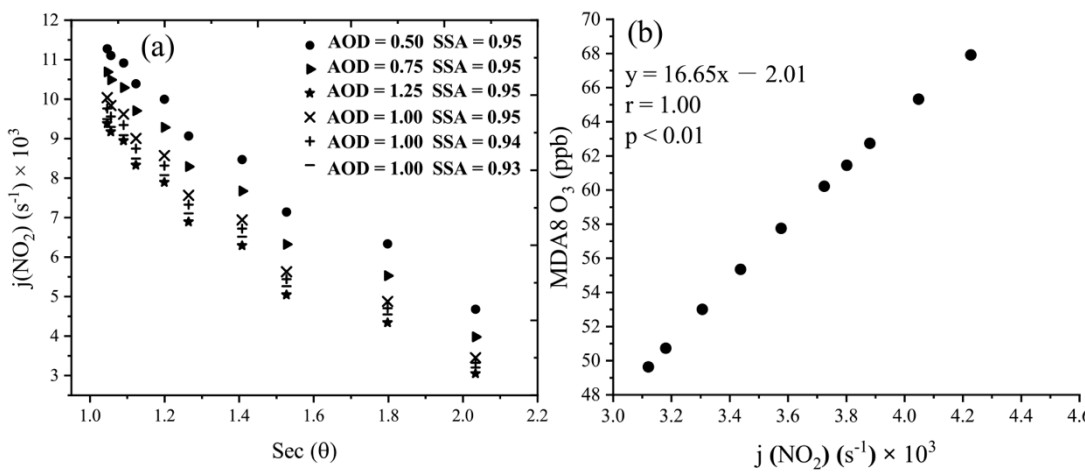

**Figure 8.** (a) response of photolysis rate of $NO_2$, $j(NO_2)$ to different values of aerosol optical depth (AOD) and single scatter factor (SSA) and (b) change in MDA8 $O_3$ with $j(NO_2)$ simulated by the MM model for the cases with SSA=0.95 and AOD varying from 0.5 to 1.25.

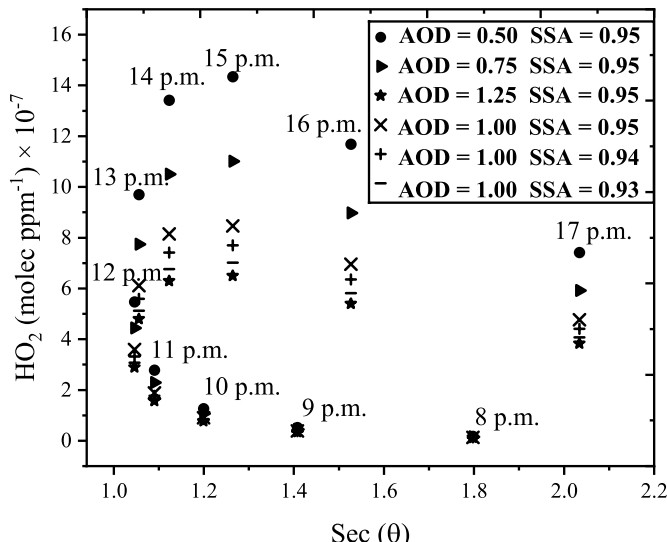

**Figure 9.** Response of concentrations of HO$_2$ to different values of aerosol optical depth (AOD) and single scatter

factor (SSA).

835