# Peer review of "Rapid increase in summer surface ozone over the North China Plain during 2013–2019: a side effect of particulate matters reduction control?"

_Atmospheric Chemistry and Physics, 2020_

## Referee Comment (RC1) · Anonymous Referee #1 · 4 Sep 2020

Overview

The paper deals with the rapid increase in summer surface ozone over the North China Plain during 2013–2019 and the hypothesis that this decrease would be a side effect of reduction control of atmospheric particulate matter is examined. I would suggest publication of the paper, after the issues raised below are addressed.

General comments

I think that there are some key measurements missing in order to undertake a proper review of this manuscript. The most important is the lack of hourly in-situ NO2 measurements from the same stations providing the hourly ozone measurements, so that

to be able to check if the ozone increase is directly related to the corresponding NO2 decrease of surface concentrations The reason is that it is already known that for most urban stations the sum NO2+O3=Ox, called also potential ozone is constant (e.g. Kley et al., 1994; Kalabokas et al., 2000), so that any NO2 decrease is directly related with an exactly equivalent increase in ozone (in ppb) through reaction with ozone (NO titration), which is very rapid. The presented data of total NO2 column might provide some indication on that, but it is expected to be less efficient than in-situ measurements.

In addition, the issue of tropospheric ozone and its subsequent influence to the boundary layer and surface background ozone concentrations should be also taken into account. In relation to that, in my opinion, a weak point of the paper is that the levels of measured surface ozone are mainly related to the photochemical ozone production over the examined region of China. On the other hand, the issue of background ozone levels and their variability within the boundary layer and the free troposphere are not (or very little) discussed. For this purpose, I think that it would be quite helpful to take into account a relatively recent extended review paper on tropospheric ozone on global scale, including SE Asia which is one of the most important global tropospheric ozone hotspots (Gaudel et al, 2018, Elem Sci Anth, 6: 39. DOI: https://doi.org/10.1525/elementa.291 and also references therein). From my perspective and based on my expertise of analyzing ozone episodes in the Mediterranean region, I would just point out that the possibility of vertical ozone transport in the troposphere influencing the boundary layer and surface ozone values (a major factor in the Mediterranean, especially in its eastern part during summer but also in its western part during spring) is not mentioned in the manuscript and so all measured ozone is considered to be produced by local photochemistry from precursor pollutant emissions emitted in China only. This might not be always the case, especially during the May-September period when the tropospheric influence to the boundary layer gets its maximum height while at the same time the tropospheric ozone maxima are observed during the same period of the year, with subsequent influence to the boundary layer and surface ozone values depending on the prevailing synoptic meteorological conditions.

In relation to the above, tropospheric vertical ozone measurements over China (e.g. Ding et al., 2008; Zhang et al., 2020) would be needed for a thorough assessment together with tropospheric satellite ozone data. In addition, synoptic weather patterns might influence greatly the tropospheric as well as the surface ozone concentrations (e.g. Kalabokas et al., 2013; Kalabokas et al., 2017) and this issue is not discussed.

Overall, I think that the submitted paper presents some interesting data and ideas regarding the recent increasing trend of surface ozone in China but I think that the above described missing information is essential for a proper review of this manuscript.

Specific comments

Page 2, lines 216-220: This is reasonable as higher NO/NO2 levels increase the ozone destruction in urban and semi urban stations, through NO titration.

Page 9, lines 235-236: This applies to stations with low NO emissions in their surroundings. As mentioned before in most urban and semi urban stations, the NO titration is the controlling factor.

Page 14, lines 379-386: This in fact reflects the preponderant role of NO titration. Lower NO emissions destroy less ozone, which in most stations is originated from the tropospheric/boundary layer background.

Supplement: I would suggest plotting also the average diurnal profiles of pollutants (O3, PM2.5) per season, at least for spring and summer.

Technical comments

Page 26, line 665 (Fig. 3): PBLH (g, h).

---

## Author Comment (AC1) · 14 Sep 2020

Dear Anonymous Referee #1,

We are submitting the point-by-point responses to your comments. We thank you for comments and suggestions, and hope you are satisfied with our responses.

The major changes made in the revised version include:

1.  Add hourly in-situ $NO_2$ measurements analysis in the supplement.

2.  Add discussions about the tropospheric $O_3$ transport and synoptic pattern effects on surface $O_3$ in the revised manuscript.

3.  Add explanation about the preponderant role of NO titration in surface ozone increase in the revised manuscript.

4.  Add the average diurnal profiles of pollutants in the revised supplement.

The 1st author: Xiaodan Ma
Corresponding author: Dr. Janping Huang

**Point-by-point responses to the Comments/Suggestions**

**Overview**

*The paper deals with the rapid increase in summer surface ozone over the North China Plain during 2013–2019 and the hypothesis that this decrease would be a side effect of reduction control of atmospheric particulate matter is examined. I would suggest publication of the paper, after the issues raised below are addressed.*

**General comments**

*1. I think that there are some key measurements missing in order to undertake a proper review of this manuscript. The most important is the lack of hourly in-situ NO2 measurements from the same stations providing the hourly ozone measurements, so that to be able to check if the ozone increase is directly related to the corresponding $NO_2$ decrease of surface concentrations The reason is that it is already known that for most urban stations the sum $NO_2+O_3=O_X$, called also potential ozone is constant (e.g. Kley et al., 1994; Kalabokas et al., 2000), so that any NO2 decrease is directly related with an exactly equivalent increase in ozone (in ppb) through reaction with ozone (NO titration), which is very rapid. The presented data of total NO2 column might provide some indication on that, but it is expected to be less efficient than in-situ measurements.*

Thank the reviewer for the suggestion. The in-situ $NO_2$ measurements are now included for a comparison between year 2013 and year 2019 (Fig.T1). The in-situ measured $NO_2$ showed a similar decreasing trend to the total column $NO_2$ (Fig. T2).

To better understand the relationship of increase in surface $O_3$ with the decrease in $NO_2$, the change in monthly mean Ox (a sum of $O_3$ and $NO_2$) is plotted in Fig. T3. It is clear that Ox showed an increasing trend over the past 7 years during daytime and nighttime in both Beijing and the NCP region. The decrease in $NO_x$ emissions could be part of the main reasons causing the rapid increase in $O_3$ over the NCP region where $O_3$ formation is dominated by the VOC-limited regime, but we do not see that "any $NO_2$ decrease is directly related with an exactly equivalent increase in ozone (in ppb) through reaction with ozone (NO titration)" as the reviewer expected. In fact, this isn't inconsistent with the statement that Ox is a conservative quantity over short time scales (Kley et al., 1994) since we are looking at the change over a long-time period (i.e., 7 years). Will add our response to address the comment in the revised version.

[Figure]

[Figure]

Figure T1. A comparison of spatial distributions of monthly mean of $NO_2$ (μg m$^{-3}$) monitored by China National Environmental Monitoring Center between (a) 2013 and (b) 2019 in eastern China (NCP indicated by the box).

[Figure]

Figure T2. Long-term changes in monthly mean of observed $NO_2$ averaged over the North China Plain in June over the period of 2013–2019.

[Figure]

Figure T3. Long-term changes in monthly mean of observed $NO_2$ averaged over the North China Plain (a) and urban areas Beijing in daytime (redline) and nighttime (blackline) in June over the period of 2013–2019.

*2.    In addition, the issue of tropospheric ozone and its subsequent influence to the boundary layer and surface background ozone concentrations should be also taken into account. In relation to that, in my opinion, a weak point of the paper is that the levels of measured surface ozone are mainly related to the photochemical ozone production over the examined region of China. On the other hand, the issue of background ozone levels and their variability within the boundary layer and the free troposphere are not (or very little) discussed. For this purpose, I think that it would be quite helpful to take into account a relatively recent extended review paper on tropospheric ozone on global scale, including SE Asia which is one of the most important global tropospheric ozone hotspots (Gaudel et al, 2018, Elem Sci Anth, 6: 39. DOI: https://doi.org/10.1525/elementa.291 and also references therein). From my perspective and based on my expertise of analyzing ozone episodes in the Mediterranean region, I would just point out that the possibility of vertical ozone transport in the troposphere influencing the boundary layer and surface ozone values (a major factor in the Mediterranean, especially in*

*its eastern part during summer but also in its western part during spring) is not mentioned in the manuscript and so all measured ozone is considered to be produced by local photochemistry from precursor pollutant emissions emitted in China only. This might not be always the case, especially during the May-September period when the tropospheric influence to the boundary layer gets its maximum height while at the same time the tropospheric ozone maxima are observed during the same period of the year, with subsequent influence to the boundary layer and surface ozone values depending on the prevailing synoptic meteorological conditions.*

We agree with the reviewer that the change in tropospheric $O_3$ could exert an important impact on ozone in the atmospheric boundary layer (ABL) or near surface. For instance, Jiang et al. (2015) presents an ozone episode in the southeast costal of China, and found that the downward transport of $O_3$ from the UTLS region driven by a typhoon is the key factor causing a large increase in surface $O_3$ by 21-42ppb. However, East Asia including the NCP region is considered as a net export region of $O_3$ from the ABL rather than an import (e.g., Cooper et al., 2010; Lin et al., 2012). On the other hand, the long-range transport of $O_3$ from Africa may exert an important impact on $O_3$ peaks in Asia around $25^0N$, i.e., the south of our study area, NCP ($32^0N$-$40^0N$) with the largest impact in boreal winter and early spring (>10 ppb) and the lowest in boreal summer (<6ppb) (Han et al., 2018).

The impacts of downward transport of tropospheric $O_3$ and regional transport are not included in our current study mainly due to two reasons. First, the MM model does not have a capability of simulating vertical transport. Second, no vertical observational data (like ozone sounding or Lidar observational data in this region over the study period) are available. We have included the first point in the Discussion and will add a statement of "the impact of tropospheric $O_3$ should be taken into account when the observational data available in this region" in the revised version to address the reviewer's comment.

*In relation to the above, tropospheric vertical ozone measurements over China (e.g. Ding et al., 2008; Zhang et al., 2020) would be needed for a thorough assessment together with tropospheric satellite ozone data. In addition, synoptic weather patterns might influence greatly the tropospheric as well as the surface ozone concentrations (e.g. Kalabokas et al., 2013; Kalabokas et al., 2017) and this issue is not discussed.*

Assessment of the change in tropospheric $O_3$ is an attractive topic, and we like to pursue it in another study separately. To our knowledge, tropospheric $O_3$ retrieved from satellite measurements remains a large uncertainty given 90% of total $O_3$ is located in the stratosphere. In addition, no ozone sounding data (e.g., Zhang et al., 2020) and other observations such as Measurement of Ozone by Airbus In-service airCraft' (MOZAIC) program data (Ding et al., 2008) are available for performing further analysis in this region during the study period. We admit that the MM model used in this study does not have a capability of quantifying the impact of vertical transport on surface $O_3$, and we have included this in our discussion.

We agree with the reviewer that synoptic pattern may exert an important impact on tropospheric and surface $O_3$ concentrations. For instance, high concentrations of surface $O_3$ or $O_3$ episodes

occurred in western Mediterranean and central Europe were usually linked with anticyclone synoptic pattern which leads to a large-scale subsidence, clear sky, and high temperature (e.g., Kalabokas et al., 2013; Kalabokas et al., 2017). Yin et al. (2019) conclude that synoptic patterns play a critical role in summer ozone pollution in eastern China. Under the control of zonally enhanced East Asian deep trough, the local hot, dry air and intense solar radiation enhance the photochemical reactions and produce more $O_3$. However, the rapid increase of surface ozone in NCP during the past seven years was mainly due to anthropogenic emissions based on recent publications (Li et al., 2019; Liu and Wang, 2020a, 2020b). The inter-annual magnitude variations of the domain synoptic patterns may have an important impact on surface $O_3$ episodes, and its impact on the long-term changes needs a further investigation. Thus, we include discussion on impact of individual meteorological factors such as air temperature, the atmospheric boundary layer height, etc. rather than synoptic patten on long-term surface $O_3$ in our study.

To address reviewer's comment, we will add additional discussions on this comment in the revised version.

*Overall, I think that the submitted paper presents some interesting data and ideas regarding the recent increasing trend of surface ozone in China but I think that the above described missing information is essential for a proper review of this manuscript.*

Thanks for positive comments and we tried our best to include all the available observational data in this study. In-site $NO_2$ measurements are included in the revised version as suggested.

**Specific comments**
*Page 2, lines 216-220: This is reasonable as higher NO/NO2 levels increase the ozone destruction in urban and semi urban stations, through NO titration.*
Thanks for comments. We agree that higher $NO/NO_2$ levels contribute to more ozone through NO titration.

*Page 9, lines 235-236: This applies to stations with low NO emissions in their surroundings. As mentioned before in most urban and semi urban stations, the NO titration is the controlling factor.*
Thanks for comments. We will add the discussion about NO titration in the manuscript.

*Page 14, lines 379-386: This in fact reflects the preponderant role of NO titration. Lower NO emissions destroy less ozone, which in most stations is originated from the tropospheric/boundary layer background.*

Thanks for your comments. We will add explanation about the preponderant role of NO titration in surface ozone increase in the revised manuscript.

*Supplement: I would suggest plotting also the average diurnal profiles of pollutants (O3, PM2.5) per season, at least for spring and summer.*

Thanks for suggestion. We will add the average diurnal profiles of pollutants in the revised supplement.

[Figure]

Figure T4. Average diurnal profiles of (a) $O_3$, (b) $NO_2$, (c) $PM_{2.5}$ in June of 2013 (black lines) and 2019 (red lines).

**Technical comments**
*Page 26, line 665 (Fig. 3): PBLH (g, h).*

Thanks. We will revise it.

**References:**

Cooper, O. R., Parrish, D. D., Stohl, A., Trainer, M., Ne ́de ́lec, P., Thouret, V., Cammas, J.-P., Oltmans, S. J., Johnson, B. J., Tarasick, D., Leblanc, T., McDermid, I. S., Jaffe, D., Gao, R., Stith, J., Ryerson, T., Aikin, K., Campos, T., Weinheimer, A., and Avery, M. A.: Increasing springtime ozone mixing ratios in the free troposphere over Western North America, Nature, 463, 344–348, doi:10.1038/nature08708, 2010.

Ding, A. J., Wang, T., Thouret, V., Cammas, J.-P., and Nédélec, P.: Tropospheric ozone climatology over Beijing: analysis of aircraft data from the MOZAIC program, Atmos. Chem. Phys., 8, 1–13, https://doi.org/10.5194/acp-8-1-2008, 2008.

Han, H., Liu, J., Yuan, H., et al. Characteristics of intercontinental transport of tropospheric ozone from Africa to Asia[J]. Atmos. Chem. Phys., 1-52. https://doi.org/10.5194/acp-2017-728, 2018.

Jiang, Y. C., Zhao, T. L., Liu, J., Xu, X. D., Tan, C. H., Cheng, X. H., Bi, X. Y., Gan, J. B., You, J. F., and Zhao, S. Z.: Why does surface ozone peak before a typhoon landing in southeast China?, Atmos. Chem. Phys., 15, 13331–13338, https://doi.org/10.5194/acp-15-13331-2015, 2015.

Kalabokas P D., et al. Examination of the atmospheric conditions associated with high and low summer ozone levels in the lower troposphere over the eastern Mediterranean. Atmos. Chem. Phys., 13:10339-10352, https://doi.org/10.5194/acp-13-10339-2013, 2013.

Kalabokas P D., et al. An investigation on the origin of regional springtime ozone episodes in the western Mediterranean. Atmos. Chem. Phys., 17, 3905–3928, https://doi.org/10.5194/acp-17-3905-2017, 2017.

Kley, Dieter & Geiss, Heiner & Mohnen, Volker. Tropospheric ozone at elevated sites and precursor emissions in the United States and Europe. Atmospheric Environment. 28. 149-158. 10.1016/1352-2310(94)90030-2, 1994.

Li, K., Jacob, D. J., Liao, H., Shen, L., and Bates, K. H.: Anthropogenic drivers of 2013-2017 trends in summer surface ozone in China, National. Acad. Sciences., 116, 422-427, https://doi.org/10.1073/pnas.1812168116, 2019

Lin, M., A. M. Fiore, O. R. Cooper, L. W. Horowitz, A. O. Langford, H. Levy II, B. J. Johnson, V. Naik, S. J. Oltmans, and C. J. Senff. Springtime high surface ozone events over the western United States: Quantifying the role of stratospheric intrusions, J. Geophys. Res., 117, D00V22, doi:10.1029/2012JD018151, 2012.

Liu, Y., Wang, T.: Worsening urban ozone pollution in China from 2013 to 2017 – Part 1: The complex and varying roles of meteorology, Atmos. Chem. Phys., 20, 6305-6321, https://doi.org/10.5194/acp- 20-6305-2020, 2020a.

Liu, Y., Wang, T.: Worsening urban ozone pollution in China from 2013 to 2017 – Part 2: The effects of emission changes and implications for multi-pollutant control, Atmos. Chem. Phys., 20, 6323-6337, https://doi.org/10.5194/acp-20-6323-2020, 2020b.

Zhang, W., Zou, Y., Zheng, X.D., Wang, N., Yan, H., Chen, Y.P., Zhao, X.J., Ji, Z.P., Li, F., Mai, B.R., Yin, C.Q., Deng, T., Fan, L.Y., Deng, X.J., Characteristics of the vertical distribution of tropospheric ozone in late autumn at Yangjiang station in Pearl River Delta (PRD), China. PartI: Observed event, Atmos. Environ., 244,117898, 10.1016/j.atmosenv. 2020.

---

## Short Comment (SC1) · 22 Sep 2020

This manuscript uses observations (including in situ and remote sensing data) and box model simulations to examine the trend of O3 over North China Plains during 2013-2019 and its impacting factors including emissions, AOD, SSA, temperature and boundary layer height (PBLH). The contribution to O3 formation from each impacting factor is quantified and found that reduction of emissions and aerosol radiative effects are the dominant factors that contribute to O3 increase from 2013 to 2019. Such analysis help understand O3 chemistry over NCP and help develop effective control strategies on reduction of ambient O3. The manuscript is written well and clearly. This

reviewer would recommend publication after addressing the following comments.

Specific comments:

1. The assumptions made in box model simulations need to be described more clearly. According to LN161, "This model computes time-dependent chemical evolution of an air parcel initialized with a known composition, assuming no additional emissions, no dilution and no heterogeneous processes", the model only takes initial concentration of chemical species, radiation and temperature and compute evolution of concentrations. It cannot directly quantify "the relative contributions of anthropogenic emissions and aerosol optical and radiative properties to the change in surface O3" stated on LN151. A few assumptions must have been made. How does the model account for different NOx and VOCs emissions, and PBLH?

LN187, Using meteorological data at a 4-hour interval appears too coarse/crude to reproduce diurnal variation of O3. Why not use hourly values?

2. The diurnal variation of O3 depends on the nighttime O3 depletion due to NO titration and dry deposition and daytime O3 formation after rush hour emissions. The box model does not account for dry deposition and additional rush hour emissions, it is a little surprising the box model can still capture the full diurnal variation of O3 in Fig. 7.

3. What is the purpose of sensitivity of jNO2 to different solar zenith angle in Figs. 8 and 9? which should not vary from 2013 to 2019. In another word, solar zenith angle is not a factor for O3 variation from 2013-2019.

LN59, TCNO2 was reported to be increased by 307% in Beijing from 1996-2011, but it decreased from 2013-2019 in Fig. 2. Are they consistent?

LN115 what is "ppb a-1"? ppb/year ?

Fig. 7b needs to be improved, different lines are hard to read.

LN97, " reducing heterogeneous uptake of reactive gases (mainly HO2 and O3), of

which the latter is more important". Box model appears more suitable to investigate such an impact.

Fig. 5g may be misleading. The positive correlation between PBLH and O3 may be simply because on those high PBLH cases, radiation and temperature are much higher, thus O3 formation is stronger. Many papers report that shallower PBL suppresses dispersion of pollutants and leads to higher O3, suggesting a negative correlation between PBLH and O3.

Sensitivity simulations summarized in Table 1 appears to attribute radiation uncertainties to aerosols (AOD and SSA). Cloud may also play critical roles, which might be the reason to explain the poor correlation between radiation and aerosol concentrations in Fig. 6b.
* * *

---

## Author Comment (AC2) · 28 Sep 2020

Dear Editor(s),

We are submitting the point-by-point responses to the reviewer's comments. We thank the reviewer for the constructive comments and suggestions, and hope you are satisfied with our responses.

The major changes in the revised version include:

1. Correct model descriptions in the manuscript.

2. Improve Fig. 7b to ensure a better readability.

3. Add discussion about radiation uncertainties to aerosols (AOD and SSA) about Fig. 6b.

The 1st author: Xiaodan Ma
Corresponding author: Dr. Janping Huang

**Point-by-point responses to the Comments/Suggestions**

*This manuscript uses observations (including in situ and remote sensing data) and box model simulations to examine the trend of $O_3$ over North China Plains during 2013-2019 and its impacting factors including emissions, AOD, SSA, temperature and boundary layer height (PBLH). The contribution to $O_3$ formation from each impacting factor is quantified and found that reduction of emissions and aerosol radiative effects are the dominant factors that contribute to $O_3$ increase from 2013 to 2019. Such analysis help understand $O_3$ chemistry over NCP and help develop effective control strategies on reduction of ambient $O_3$. The manuscript is written well and clearly. This reviewer would recommend publication after addressing the following comments.*

*Specific comments:*

*1. The assumptions made in box model simulations need to be described more clearly. According to LN161, "This model computes time-dependent chemical evolution of an air parcel initialized with a known composition, assuming no additional emissions, no dilution and no heterogeneous processes", the model only takes initial concentration of chemical species, radiation and temperature and compute evolution of concentrations. It cannot directly quantify "the relative contributions of anthropogenic emissions and aerosol optical and radiative properties to the change in surface $O_3$" stated on LN151. A few assumptions must have been made. How does the model account for different NOx and VOCs emissions, and PBLH?*

Thanks for pointing out the problem. Indeed, something was mixed up in the original description. NOx and VOCs emissions were changed to assess the impact of emissions on surface $O_3$ simulations (see Table 1). It is assumed that no dilution is included in the simulations given the difficulty of getting inputs to calculate the dilution rate. Dry deposition is included in all the simulations and sensitivity runs. PBLH was ingested as the model requests every 4 hours to support the calculation of entrainment term. The issue is corrected and can be found in the revised version.

*LN187, using meteorological data at a 4-hour interval appears too coarse/crude to reproduce diurnal variation of $O_3$. Why not use hourly values?*

Yes, theoretically, hourly input data could reproduce more reasonable diurnal variations in surface $O_3$ simulations. However, according to our tests, the computational time increased substantially when input data were ingested hourly. Meanwhile, the MM model was able to capture the diurnal variation pattern reasonably when the input data were ingested every 4 hours. Thus, we decided to ingest the input data every 4 hours for all the numerical experiments.

*2. The diurnal variation of $O_3$ depends on the nighttime $O_3$ depletion due to NO titration and dry deposition and daytime $O_3$ formation after rush hour emissions. The box model does not account for dry deposition and additional rush hour emissions. It is a little surprising the box model can still capture the full diurnal variation of $O_3$ in Fig. 7.*

Sorry for the confused. Please see our response to the 1st comment provided above. Dry deposition was included in our simulations. Such as information is included in the revised version.

With the emission inputs at an interval of 4 hours, the diurnal variation in surface $O_3$ was reproduced.

*3. What is the purpose of sensitivity of $jNO_2$ to different solar zenith angle in Figs. 8 and 9? which should not vary from 2013 to 2019. In another word, solar zenith angle is not a factor for $O_3$ variation from 2013-2019.*

We agree with the reviewer that the solar zenith angle is not a factor in driving the increase in surface $O_3$ from 2013 to 2019. The main purpose of Figures 8 and 9 is to examine the sensitivity of $jNO_2$, the daily 8-hr maximum $O_3$ and $HO_2$ radicals to the changes in AOD and SSA. Solar zenith angle is included in the figures for a comparison since both $jNO_2$ and $HO_2$ radicals are very sensitive to changes in solar zenith angle but show very different change trend with varying solar zenith angle. For instance, $jNO_2$ shows a nearly linear change with Sec ($\theta$) wheras $HO_2$ doesn't. On the other hand, $jNO_2$ shows the largest sensitivity to AOD and SSA at noontime while $HO_2$ has a largest sensitivity around 3:00 pm local standard time which is consistent with the peak hour of surface $O_3$.

*LN59, $TCNO_2$ was reported to be increased by 307% in Beijing from 1996-2011, but it decreased from 2013-2019 in Fig. 2. Are they consistent?*

Yes, they are correct. The change trend was reversed around 2013 since a strict NOx emissions control measures were implemented in China (e.g., Huang et al., 2013; Zeng et al., 2019).

*LN115 what is "ppb a-1"? ppb/year?*

Yes. "ppb a-1" represent ppb per annual (year). It is corrected.

*Fig. 7b needs to be improved, different lines are hard to read.*

Thanks for the suggestion, the figure is replotted with better color identification to improve readability.

[Figure]

*LN97, " reducing heterogeneous uptake of reactive gases (mainly HO$_2$ and O$_3$), of which the latter is more important". Box model appears more suitable to investigate such an impact.*

Yes, the current box model needs to include a detailed aerosol chemistry and observation-based uptake coefficients to achieve the target. We added this in the revised version.

*Fig. 5g may be misleading. The positive correlation between PBLH and O$_3$ may be simply because on those high PBLH cases, radiation and temperature are much higher, thus O$_3$ formation is stronger. Many papers report that shallower PBL suppresses dispersion of pollutants and leads to higher O$_3$, suggesting a negative correlation between PBLH and O$_3$.*

Yes, we agree. This could represent one of the cases when radiation is strong, temperature is higher while the PBL height is higher either. Higher height of the PBL also could lead to the mixing of near surface air with the O$_3$ rich air aloft, resulting in the observed enhancements in surface O$_3$ (REDDY et al., 2012). O$_3$ formation could be suppressed by the a shallow PBL due to the NO titration. We have added an additional statement "whereas some other studies reported that a shallow PBL suppresses the dispersion of pollutants and leads to higher O$_3$, suggesting a negative correlation between PBLH and O$_3$ (Yan et al., 2018; Jiang et al., 2016; Wei et al., 2016; Huang et al., 2005)". Please see the revised version.

*Sensitivity simulations summarized in Table 1 appears to attribute radiation uncertainties to aerosols (AOD and SSA). Cloud may also play critical roles, which might be the reason to explain the poor correlation between radiation and aerosol concentrations in Fig. 6b.*

Thanks for the comment. We have added the statement with cloud may also play a critical role, which might be another reason to explain the poor correlation between radiation and aerosol concentrations in Fig. 6b" in the revised version.

**References:**

Jiang, Y. C., Zhao, T. L., Liu, J., Xu, X. D., Tan, C. H., Cheng, X. H., Bi, X. Y., Gan, J. B., You, J. F., and Zhao, S. Z.: Why does surface ozone peak before a typhoon landing in southeast China?, Atmos. Chem. Phys., 15, 13331–13338, https://doi.org/10.5194/acp-15-13331-2015, 2015.

Huang, J., Zhou, C., Lee, X., Bao, Y., Zhao, X., Fung, J., RICHTER, Andreas, Liu, X., and Zheng, Y.: The effects of rapid urbanization on the levels in tropospheric nitrogen dioxide and ozone over East China, Atmos. Environ., 77, 558-567, https://doi.org/10.1016/j.atmosenv.2013.05.030, 2013.

Huang, Jian-Ping & Fung, Jimmy & Lau, Alexis & Qin, Yu. (2005). Numerical simulation and process analysis of typhoon-related ozone episodes in Hong Kong. Journal of Geophysical Research. 110. 10.1029/2004JD004914.

REDDY, K.K., NAJA, M., OJHA, N. et al. Influences of the boundary layer evolution on surface ozone variations at a tropical rural site in India. J Earth Syst Sci 121, 911–922 (2012). https://doi.org/10.1007/s12040-012-0200-z.

Wei, Xiaolin & Lam, Ka-se & Cao, Chunyan & Li, Hui & He, Jiajia. (2016). Dynamics of the Typhoon Haitang Related High Ozone Episode over Hong Kong. Advances in Meteorology. 2016. 1-12. 10.1155/2016/6089154.

Yan R C,Ye H,Lin X,et al. 2018.Characteristics and influence factors of ozone pollution in Hangzhou[J].Acta Scientiae Circumstantiae,38( 3) : 1128-1136

Zeng, Y., Cao, Y., Qiao, X., Seyler, B. C., and Tang, Y.: Air pollution reduction in China: Recent success but great challenge for the future, Sci. Total. Environ., 663, 329-337, https://doi.org/10.1016/j.scitotenv.2019.01.262, 2019.

---

## Referee Comment (RC2) · Guy Brasseur (Referee) · 29 Sep 2020

The paper in its present form is clearly written and could be published after revisions

At this stage of the review process, I would like to add a few points and suggestions to be carefully addressed by the authors:

1. Since the focus of the paper is on the response of ozone to possible forcing processes, I wonder why so little is said about the role of heterogeneous chemistry. There is growing evidence that the increase in ozone is related to the increase in HO2 due to the reduced scavenging of HO2 by aerosols (PM) more than a change in the J-values.

[Figure]

The paper highlights the change in the J(NO2) due reduced PM concentrations, but does not provide the quantitative response associated with the change in heterogenous processes. It would be important to discuss this aspect, even if no specific simulation of this effect has been done. 2. A more convincing discussion must be provided regarding the role of the boundary layer, the changing meteorology (e.g., average cloudiness, precipitation, etc,) and the surface deposition processes. Some of these may not be explicitly treated in a box model, but should be discussed with appropriate references. 3. There should be a discussion about the processes that have changed HOx chemistry (which affects the ozone production and loss) and this includes, for example, the HONO and formaldehyde photolysis.

---

## Author Comment (AC3) · 14 Oct 2020

Dear Guy Brasseur Referee:

We are submitting the point-by-point responses to your comments. We thank you for comments and suggestions, and hope you are satisfied with our responses.

The major changes made in the revised version include:

1.  Added related discussion about the role of heterogeneous chemistry in the revised manuscript.
2.  Added more discussions on the role of the boundary layer and changing meteorology in $O_3$ trends.
3.  Added a discussion on the HOx chemical source in the revised manuscript.

On behalf of the co-authors,

First Author: Xiaodan Ma
Corresponding Author: Jianping Huang

**Point-by-point responses to the Comments/Suggestions of Reviewer #3**

**Overview**

*The paper in its present form is clearly written and could be published after revisions. At this stage of the review process, I would like to add a few points and suggestions to be carefully addressed by the authors:*

**General comments**

*1. Since the focus of the paper is on the response of ozone to possible forcing processes, I wonder why so little is said about the role of heterogeneous chemistry. There is growing evidence that the increase in ozone is related to the increase in $HO_2$ due to the reduced scavenging of $HO_2$ by aerosols (PM) more than a change in the J-values. The paper highlights the change in the $J(NO_2)$ due reduced PM concentrations, but does not provide the quantitative response associated with the change in heterogenous processes. It would be important to discuss this aspect, even if no specific simulation of this effect has been done.*

Thanks for the reviewer's suggestion. According to recent researches, decrease in $PM_{2.5}$ was considered as one of the important causes leading to such an increase in surface $O_3$ mainly due to additional $O_3$ production associated with reduced sink of hydroperoxyl radicals ($HO_2$) (Li et al., 2019). They pointed out that increase in surface $O_3$ associated with decrease in $PM_{2.5}$ was more prominent than that with reduction of NOx emissions over the NCP region where $O_3$ formation was dominated by VOC-limited regime. Liu and Wang (2020a, 2020b) found the reduction of PM emissions increased the $O_3$ levels by enhancing the photolysis rates and reducing heterogeneous uptake of reactive gases (mainly $HO_2$ and $O_3$), of which the latter is more important than the former. However, the MM model does not include aqueous-phase chemistry that has been implemented in the 3D meteorology/chemistry models (e.g., Li et al., 2019; Liu and Wang, 2020a, 2020b). Thus, inclusion of detailed aerosol chemistry and observation-based uptake coefficients in a box model like MM is necessary to provide more accurate assessment of impact of aerosol radiative effect on surface $O_3$ change in the future.

We have included the related discussion about the role of heterogeneous chemistry both in the Introduction and Discussion part (See L92-98 and L407-413).

*2. A more convincing discussion must be provided regarding the role of the boundary layer, the changing meteorology (e.g., average cloudiness, precipitation, etc.,) and the surface deposition processes. Some of these may not be explicitly treated in a box model, but should be discussed with appropriate references.*

Thanks for the comment. We agree with the reviewer that the boundary layer and the changes in other meteorological factors such as cloudiness, precipitation etc. played an important role in increase in surface $O_3$. In fact, we have conducted two cases (cases F and G in Table 1) to assess the impact of the planetary boundary layer height (PBLH) on the change in surface $O_3$. In addition, we have included two cases (i.e., E and G in Table 1) to investigate the impact of surface maximum air temperature on increase in surface $O_3$. To highlight the reviewer's concern with the role of changing meteorology, we cited a relevant reference on this topic, "The

influence of changing meteorological factors on the change trend in surface $O_3$ may vary greatly with regions and time. In addition to air temperature and the boundary layer conditions, other meteorological factors such as cloud cover, precipitation, wind fields played an important role in driving the changes in surface $O_3$ observed in many places of China (e.g., Liu and Wang, 2020a)" (see L416-420).

*3. There should be a discussion about the processes that have changed $HO_x$ chemistry (which affects the ozone production and loss) and this includes, for example, the HONO and formaldehyde photolysis.*

We agree with the reviewer that $HO_x$ chemistry is an important factor affecting the production and loss of $O_3$. Now a statement with "The HONO photolysis as the primary production of OH radicals and the formaldehyde (HCHO) photolysis as the net radical source of $HO_2$ can lead to major changes in the HOx and NOx budget that may have an important effect on $O_3$ production and loss (e.g., Aumont et al., 2003; Brasseur et al., 2006; Lin et al., 2012)" is added on the revised version (see L174-179).

**References:**

Aumont, B., Chervier, F., and Laval, S.: Contribution of HONO sources to the NOx/HOx/O3 chemistry in the polluted boundary layer, Atmos. Environ., 37, 487-498, https://doi.org/10.1016/S1352-2310(02)00920-2, 2003.

Brasseur, G. P., and Solomon, S.: Aeronomy of the middle atmosphere: Chemistry and physics of the stratosphere and mesosphere, Springer Science & Business Media, 2006.

Li, K., Jacob, D. J., Liao, H., Shen, L., and Bates, K. H.: Anthropogenic drivers of 2013-2017 trends in summer surface ozone in China, National. Acad. Sciences., 116, 422-427, https://doi.org/10.1073/pnas.1812168116, 2019.

Lin, Y. C., Schwab, J., Demerjian, K., Bae, M.-S., Chen, W.-N., Sun, Y., Zhang, q., Hung, H.-M., and Perry, J.: Summertime formaldehyde observations in New York City: Ambient levels, sources and its contribution to HOx radicals, J. Geophys. Res., 117, D08305, https://doi.org/10.1029/2011JD016504, 2012.

Liu, Y., and Wang, T.: Worsening urban ozone pollution in China from 2013 to 2017 – Part 2: The effects of emission changes and implications for multi-pollutant control, Atmos. Chem. Phys. Discuss., 2020, 1-27, 10.5194/acp-2020-53, 2020a.

Liu, Y., and Wang, T.: Worsening urban ozone pollution in China from 2013 to 2017 – Part 1: The complex and varying roles of meteorology, Atmos. Chem. Phys., 20, 6305-6321, https://doi.org/10.5194/acp-20-6305-2020, 2020b.

---

## Author Response (AR1)

Dear Editors:

We are submitting the revised manuscript acp-2020-385 together with point-by-point responses
to the reviewer' comments. We appreciate the reviewers for the valuable comments. We hope
you and the reviewers are satisfied with our responses and revision.

The major changes made in the revised version include:

1. Added hourly in-situ $NO_2$ measurement analyses in the supplement.
2. Added additional discussions about the tropospheric $O_3$ transport and synoptic pattern
   effects on surface $O_3$ in the revised manuscript.
3. Added explanations about the preponderant role of NO titration in surface ozone increase
   in the revised manuscript.
4. Added the average diurnal profiles of pollutants in the revised supplement.
5. Increased a detailed model description in the manuscript.
6. Replotted Fig. 7b to ensure a better readability.
7. Added discussion about radiation uncertainties to aerosols (AOD and SSA) about Fig. 6b.
8. Added more discussions on the role of the boundary layer and changing meteorology in $O_3$
   trends.
9. Added a discussion on the HOx chemical source in the revised manuscript.

We look forward to hearing from you regarding our revision soon.

On behalf of the co-authors,

First Author: Xiaodan Ma
Corresponding Author: Jianping Huang

**Point-by-point responses to the Comments/Suggestions of Reviewer #1**
**Overview**

*The paper deals with the rapid increase in summer surface ozone over the North China Plain during 2013–2019 and the hypothesis that this decrease would be a side effect of reduction control of atmospheric particulate matter is examined. I would suggest publication of the paper, after the issues raised below are addressed.*

**General comments**

*1. I think that there are some key measurements missing in order to undertake a proper review of this manuscript. The most important is the lack of hourly in-situ $NO_2$ measurements from the same stations providing the hourly ozone measurements, so that to be able to check if the ozone increase is directly related to the corresponding $NO_2$ decrease of surface concentrations The reason is that it is already known that for most urban stations the sum $NO_2+O_3=O_X$, called also potential ozone is constant (e.g. Kley et al., 1994; Kalabokas et al., 2000), so that any $NO_2$ decrease is directly related with an exactly equivalent increase in ozone (in ppb) through reaction with ozone (NO titration), which is very rapid. The presented data of total $NO_2$ column might provide some indication on that, but it is expected to be less efficient than in-situ measurements.*

Thank the reviewer for the suggestion. The in-situ $NO_2$ measurements are now included in supplementary material for a comparison between year 2013 and year 2019 (See Fig.S4 in the supplementary material and Fig.T1 attached with this letter). The in-situ measured $NO_2$ showed a similar decreasing trend to the total column $NO_2$ (see Fig.S5 in the supplementary material and Fig. T2 with this letter).

To better understand the relationship of increase in surface $O_3$ with the decrease in $NO_2$, the change in monthly mean Ox (a sum of $O_3$ and $NO_2$) is plotted in Fig. T3 (see Fig.S2 in the supplementary material). It is clear that Ox showed an increasing trend over the past 7 years during daytime and nighttime in both Beijing and the NCP region. The decrease in $NO_x$ emissions could be part of the main reasons causing the rapid increase in $O_3$ over the NCP region where $O_3$ formation is dominated by the VOC-limited regime, but we do not see that "any $NO_2$ decrease is directly related with an exactly equivalent increase in ozone (in ppb) through reaction with ozone (NO titration)" as the reviewer expected. In fact, this is consistent with the statement that Ox is a conservative quantity (Kley et al., 1994) since we are looking at the changes over a long-time period (i.e., 7 years) rather than a short time period.

[Figure]

Figure T1. A comparison of spatial distributions of monthly mean of $NO_2$ ($\mu g\ m^{-3}$) monitored by China National Environmental Monitoring Center between (a) 2013 and (b) 2019 in eastern China (NCP indicated by the box).

[Figure]

$y = -1.77x + 37.16$
$R^2 = 0.8856$
$p < 0.05$

75 Figure T2. Long-term changes in monthly mean of observed $NO_2$ averaged over the North China Plain in June over the period of 2013–2019.

[Figure]

Figure T3. Long-term changes in monthly mean of observed Ox ($NO_2 + O_3$) averaged over the North China Plain (a) and urban areas Beijing in daytime (redline) and nighttime (blackline) in June over the period of 2013–2019.

80

2. *In addition, the issue of tropospheric ozone and its subsequent influence to the boundary layer and surface background ozone concentrations should be also taken into account. In*

*relation to that, in my opinion, a weak point of the paper is that the levels of measured surface ozone are mainly related to the photochemical ozone production over the examined region of*

85 *China. On the other hand, the issue of background ozone levels and their variability within the boundary layer and the free troposphere are not (or very little) discussed. For this purpose, I think that it would be quite helpful to take into account a relatively recent extended review paper on tropospheric ozone on global scale, including SE Asia which is one of the most important global tropospheric ozone hotspots (Gaudel et al, 2018, Elem Sci Anth, 6: 39. DOI:*

90 *https://doi.org/10.1525/elementa.291 and also references therein). From my perspective and based on my expertise of analyzing ozone episodes in the Mediterranean region, I would just point out that the possibility of vertical ozone transport in the troposphere influencing the boundary layer and surface ozone values (a major factor in the Mediterranean, especially in its eastern part during summer but also in its western part during spring) is not mentioned in*

95 *the manuscript and so all measured ozone is considered to be produced by local photochemistry from precursor pollutant emissions emitted in China only. This might not be always the case, especially during the May-September period when the tropospheric influence to the boundary layer gets its maximum height while at the same time the tropospheric ozone maxima are observed during the same period of the year, with subsequent influence to the boundary layer*

100 *and surface ozone values depending on the prevailing synoptic meteorological conditions.*

We agree with the reviewer that the change in tropospheric $O_3$ could exert an important impact on ozone in the atmospheric boundary layer (ABL) or near surface. For instance, Jiang et al. (2015) presents an ozone episode in the southeast costal of China, and found that the downward

105 transport of $O_3$ from the UTLS region driven by a typhoon is the key factor causing a large increase in surface $O_3$ by 21-42 ppb. In addition, East Asia including the NCP region is considered as a net export region of $O_3$ from the ABL rather than an import (e.g., Cooper et al., 2010; Lin et al., 2012a). The long-range transport of $O_3$ from Africa may exert an important impact on $O_3$ peaks in Asia around $25^0N$, i.e., the south of our study area, NCP ($32^0N$-$40^0N$)

110 with the largest impact in boreal winter and early spring (>10 ppb) and the lowest in boreal summer (<6ppb) (Han et al., 2018; Gaudel et al, 2018).

The impacts of downward transport of tropospheric $O_3$ and regional transport are not included in our current study mainly due to two reasons. First, the MM model does not have a capability

115 of simulating vertical transport. Second, no vertical observational data (like ozone sounding or Lidar observational data in this region over the study period) are available.

However, according to the reviewer's suggestion, we have added additional discussions on background ozone levels and their variability within the boundary layer and the free troposphere

120 and their impact on the increase in surface $O_3$. The seasonal distributions of $O_3$ in the upper troposphere (UT) show a summer maximum that coincides with the maximum photochemical activity in the North Hemisphere (Gaudel et al, 2018). A broad spring/summer ozone enhancement across northern mid-latitudes with a band of enhanced summertime ozone stretching from North Africa, across the Mediterranean Sea to East Asia at 5–7 km and 7–9 km

125 has been detected (Worden et al., 2009). Under favorable synoptic patterns, the high $O_3$ in the UT may exert an important impact on surface $O_3$ concentrations (e.g., Kalabokas et al., 2013;

Kalabokas et al., 2017).

Added discussion can be found in L433-443 of the revised manuscript.

*In relation to the above, tropospheric vertical ozone measurements over China (e.g. Ding et al., 2008; Zhang et al., 2020) would be needed for a thorough assessment together with tropospheric satellite ozone data. In addition, synoptic weather patterns might influence greatly the tropospheric as well as the surface ozone concentrations (e.g. Kalabokas et al., 2013; Kalabokas et al., 2017) and this issue is not discussed.*

Assessment of the change in tropospheric $O_3$ is an attractive topic, and we like to pursue it in another study separately. To our knowledge, tropospheric $O_3$ retrieved from satellite measurements remains a large uncertainty given 90% of total $O_3$ is located in the stratosphere. In addition, no ozone sounding data (e.g., Zhang et al., 2020) and other observations such as Measurement of Ozone by Airbus In-service airCraft' (MOZAIC) program data (Ding et al., 2008) are available for performing further analysis in this region during the study period. We admit that the MM model used in this study does not have a capability of quantifying the impact of vertical transport on surface $O_3$, and we have included this issue in our discussion. "The impact of tropospheric $O_3$ should be taken into account when the observational data available in NCP region." (see L441-443).

We agree with the reviewer that synoptic pattern may exert an important impact on tropospheric and surface $O_3$ concentrations. For instance, high concentrations of surface $O_3$ or $O_3$ episodes occurred in western Mediterranean and central Europe were usually linked with anticyclone synoptic pattern which led to a large-scale subsidence, clear sky, and high temperature (e.g., Kalabokas et al., 2013; Kalabokas et al., 2017). Yin et al. (2019) concluded that synoptic patterns played a critical role in summer ozone pollution in eastern China. Under the control of zonally enhanced East Asian deep trough, the local hot, dry air and intense solar radiation enhanced the photochemical reactions and produced more $O_3$. The inter-annual magnitude variations of the domain synoptic patterns may have an important impact on surface $O_3$ episodes, and its impact on the long-term changes needs a further investigation. Thus, we include discussion on impact of individual meteorological factors such as air temperature, the atmospheric boundary layer height, etc. rather than synoptic patten on long-term surface $O_3$ in our study.

To address reviewer's comment, we added our discussions on this comment in the revised version (see L444-452 in the revised version).

*Overall, I think that the submitted paper presents some interesting data and ideas regarding the recent increasing trend of surface ozone in China but I think that the above described missing information is essential for a proper review of this manuscript.*

Thanks for positive comments and we tried our best to include all the available observational data in this study. In-site $NO_2$ measurements are included in the revised version as suggested.

**Specific comments**

*Page 2, lines 216-220: This is reasonable as higher NO/NO2 levels increase the ozone destruction in urban and semi urban stations, through NO titration.*

Thanks for comments. We agree that higher $NO/NO_2$ levels contribute to more ozone through NO titration.

*Page 9, lines 235-236: This applies to stations with low NO emissions in their surroundings. As mentioned before in most urban and semi urban stations, the NO titration is the controlling factor.*

Thanks for comments. We have added the discussion on NO titration in the manuscript.

*Page 14, lines 379-386: This in fact reflects the preponderant role of NO titration. Lower NO emissions destroy less ozone, which in most stations is originated from the tropospheric/boundary layer background.*

Thanks for your comments. We added additional explanation on the preponderant role of NO titration in surface ozone increase in the revised manuscript (see L402-403).

*Supplement: I would suggest plotting also the average diurnal profiles of pollutants (O3, PM2.5) per season, at least for spring and summer.*

Thanks for suggestion. We added the average diurnal profiles of pollutants in the revised supplement (See Fig.S6 in the supplementary material).

[Figure]

[Figure]

[Figure]

Figure T4. Average diurnal profiles of (a) $O_3$, (b) $NO_2$, (c) $PM_{2.5}$ in June of 2013 (black lines) and 2019 (red lines).

**Technical comments**
*Page 26, line 665 (Fig. 3): PBLH (g, h).*

Thanks. It's revised.

**Point-by-point responses to the Comments/Suggestions of Reviewer #2**

*This manuscript uses observations (including in situ and remote sensing data) and box model simulations to examine the trend of $O_3$ over North China Plains during 2013-2019 and its impacting factors including emissions, AOD, SSA, temperature and boundary layer height (PBLH). The contribution to $O_3$ formation from each impacting factor is quantified and found that reduction of emissions and aerosol radiative effects are the dominant factors that contribute to $O_3$ increase from 2013 to 2019. Such analysis help understand $O_3$ chemistry over NCP and help develop effective control strategies on reduction of ambient $O_3$. The manuscript is written well and clearly. This reviewer would recommend publication after addressing the following comments.*

***Specific comments:***

*1. The assumptions made in box model simulations need to be described more clearly. According to LN161, "This model computes time-dependent chemical evolution of an air parcel initialized with a known composition, assuming no additional emissions, no dilution and no heterogeneous processes", the model only takes initial concentration of chemical species, radiation and temperature and compute evolution of concentrations. It cannot directly quantify "the relative contributions of anthropogenic emissions and aerosol optical and radiative properties to the change in surface $O_3$" stated on LN151. A few assumptions must have been made. How does the model account for different NOx and VOCs emissions, and PBLH?*

Thanks for pointing out the problem. Indeed, something was mixed up in the original description. NOx and VOCs emissions were changed to assess the impact of emissions on surface $O_3$ simulations (see Table 1). Dry deposition is included in all the simulations and sensitivity runs. The PBLH was ingested as the model requests every 4 hours to support the calculation of entrainment term.

The issue is corrected and can be found in the revised version. "This model computes time-dependent chemical evolution of an air parcel initialized with a known composition and additional emissions. It is assumed that no dilution is included in the simulations given the difficulty of getting inputs to calculate the dilution rate. The transport in and out of air pollutants reached a quasi-equilibrium state over the study domain and no heterogeneous processes was included in the MM model." (see L161-165).

*LN187, using meteorological data at a 4-hour interval appears too coarse/crude to reproduce diurnal variation of $O_3$. Why not use hourly values?*

Yes, theoretically, hourly input data could reproduce more reasonable diurnal variations in surface $O_3$ simulations. However, according to our tests, the computational time increased substantially when input data were ingested hourly. Meanwhile, the MM model was able to capture the diurnal variation pattern reasonably when the input data were ingested every 4 hours. Thus, we decided to ingest the input data every 4 hours for all the numerical simulations and

sensitivity studies.

*2. The diurnal variation of $O_3$ depends on the nighttime $O_3$ depletion due to NO titration and dry deposition and daytime $O_3$ formation after rush hour emissions. The box model does not account for dry deposition and additional rush hour emissions. It is a little surprising the box model can still capture the full diurnal variation of $O_3$ in Fig. 7.*

Sorry for the confused. Dry deposition, time-varying meteorological inputs and emission drivers were included in the MM model. With such settings, the MM model was able to capture the diurnal variation in surface $O_3$. Such an information is included in the revised version (see L160-165 and L183-193).

*3. What is the purpose of sensitivity of $jNO_2$ to different solar zenith angle in Figs. 8 and 9? which should not vary from 2013 to 2019. In another word, solar zenith angle is not a factor for $O_3$ variation from 2013-2019.*

The solar zenith angle is not a factor in driving the increase in surface $O_3$ from 2013 to 2019. The main purpose of Figures 8 and 9 is to examine the sensitivity of $jNO_2$, the daily 8-hr maximum $O_3$ and $HO_2$ radicals to the changes in AOD and SSA. The solar zenith angle is included in the figures for a comparison since both $jNO_2$ and $HO_2$ radicals are very sensitive to changes in solar zenith angle but show very different change trend with varying solar zenith angle. For instance, $jNO_2$ shows a nearly linear change with Sec ($\theta$) wheras $HO_2$ doesn't. On the other hand, $jNO_2$ shows the largest sensitivity to AOD and SSA at noontime while $HO_2$ has a largest sensitivity around 3:00 pm local standard time which is consistent with the peak hour of surface $O_3$.

*LN59, $TCNO_2$ was reported to be increased by 307% in Beijing from 1996-2011, but it decreased from 2013-2019 in Fig. 2. Are they consistent?*

Yes, they are correct. 2013 was the transition year. The change trend was reversed around 2013 since a strict NOx emissions control measures were implemented in China (e.g., Huang et al., 2013; Zeng et al., 2019).

*LN115 what is "ppb a-1"? ppb/year?*

Yes. "ppb a-1" represent ppb per annual (year). Corrected.

*Fig. 7b needs to be improved, different lines are hard to read.*

Thanks for the suggestion, the figure is replotted with better color identification to improve readability.

[Figure]

*LN97, " reducing heterogeneous uptake of reactive gases (mainly HO₂ and O₃), of which the latter is more important". Box model appears more suitable to investigate such an impact.*

Yes, the current box model needs to include a detailed aerosol chemistry and observation-based uptake coefficients to achieve the target. We added this in the revised version (see L412-413).

*Fig. 5g may be misleading. The positive correlation between PBLH and O₃ may be simply because on those high PBLH cases, radiation and temperature are much higher, thus O₃ formation is stronger. Many papers report that shallower PBL suppresses dispersion of pollutants and leads to higher O₃, suggesting a negative correlation between PBLH and O₃.*

We agree. This could represent one of the cases when radiation is strong, temperature is higher while the PBL height is higher either. Higher height of the PBL also could lead to the mixing of near surface air with the O₃ rich air aloft, resulting in the observed enhancements in surface O₃ (Reddy et al., 2012). O₃ formation could be suppressed by the a shallow PBL due to the NO titration. We have added an additional statement "On the other hand, some studies found a negative correlation between the PBLH and O₃. They claimed that a shallower PBL may suppress the dispersion of pollutants and lead to higher O₃ (Yan et al., 2018; Jiang et al., 2016; Wei et al., 2016; Huang et al., 2005)". Please see the revised version (in L304-310).

*Sensitivity simulations summarized in Table 1 appears to attribute radiation uncertainties to aerosols (AOD and SSA). Cloud may also play critical roles, which might be the reason to explain the poor correlation between radiation and aerosol concentrations in Fig. 6b.*

Thanks for the comment. We have added the statement with "cloud may also play a critical role, which might be another reason to explain the poor correlation between radiation and aerosol concentrations in Fig. 6b" in the revised version.

**Point-by-point responses to the Comments/Suggestions of Reviewer #3**
**Overview**

*The paper in its present form is clearly written and could be published after revisions. At this stage of the review process, I would like to add a few points and suggestions to be carefully addressed by the authors:*

**General comments**

*1. Since the focus of the paper is on the response of ozone to possible forcing processes, I wonder why so little is said about the role of heterogeneous chemistry. There is growing evidence that the increase in ozone is related to the increase in $HO_2$ due to the reduced scavenging of $HO_2$ by aerosols (PM) more than a change in the J-values. The paper highlights the change in the $J(NO_2)$ due reduced PM concentrations, but does not provide the quantitative response associated with the change in heterogenous processes. It would be important to discuss this aspect, even if no specific simulation of this effect has been done.*

Thanks for the reviewer's suggestion. According to recent researches, decrease in $PM_{2.5}$ was considered as one of the important causes leading to such an increase in surface $O_3$ mainly due to additional $O_3$ production associated with reduced sink of hydroperoxyl radicals ($HO_2$) (Li et al., 2019a). They pointed out that increase in surface $O_3$ associated with decrease in $PM_{2.5}$ was more prominent than that with reduction of NOx emissions over the NCP region where $O_3$ formation was dominated by VOC-limited regime. Liu and Wang (2020a, 2020b) found the reduction of PM emissions increased the $O_3$ levels by enhancing the photolysis rates and reducing heterogeneous uptake of reactive gases (mainly $HO_2$ and $O_3$), of which the latter is more important than the former. However, the MM model does not include aqueous-phase chemistry that has been implemented in the 3D meteorology/chemistry models (e.g., Li et al., 2019a; Liu and Wang, 2020a, 2020b). Thus, inclusion of detailed aerosol chemistry and observation-based uptake coefficients in a box model like MM is necessary to provide more accurate assessment of impact of aerosol radiative effect on surface $O_3$ change in the future.

We have included the related discussion about the role of heterogeneous chemistry both in the Introduction and Discussion part (See L92-98 and L407-413).

*2. A more convincing discussion must be provided regarding the role of the boundary layer, the changing meteorology (e.g., average cloudiness, precipitation, etc.,) and the surface deposition processes. Some of these may not be explicitly treated in a box model, but should be discussed with appropriate references.*

Thanks for the comment. We agree with the reviewer that the boundary layer and the changes in other meteorological factors such as cloudiness, precipitation etc. played an important role in increase in surface $O_3$. In fact, we have conducted two cases (cases F and G in Table 1) to assess the impact of the planetary boundary layer height (PBLH) on the change in surface $O_3$. In addition, we have included two cases (i.e., E and G in Table 1) to investigate the impact of surface maximum air temperature on increase in surface $O_3$. To highlight the reviewer's concern with the role of changing meteorology, we cited a relevant reference on this topic, "The

influence of changing meteorological factors on the change trend in surface $O_3$ may vary greatly with regions and time. In addition to air temperature and the boundary layer conditions, other meteorological factors such as cloud cover, precipitation, wind fields played an important role in driving the changes in surface $O_3$ observed in many places of China (e.g., Liu and Wang, 2020a)" (see L416-420).

*3. There should be a discussion about the processes that have changed $HO_x$ chemistry (which affects the ozone production and loss) and this includes, for example, the HONO and formaldehyde photolysis.*

We agree with the reviewer that $HO_x$ chemistry is an important factor affecting the production and loss of $O_3$. Now a statement with "The HONO photolysis as the primary production of OH radicals and the formaldehyde (HCHO) photolysis as the net radical source of $HO_2$ can lead to major changes in the HOx and NOx budget that may have an important effect on $O_3$ production and loss (e.g., Aumont et al., 2003; Brasseur et al., 2006; Lin et al., 2012b)" is added on the revised version (see L174-179).

**References:**

[revised manuscript text omitted]